# The doodle dilemma: How the physical health of 'Designer-crossbreed' Cockapoo, Labradoodle and Cavapoo dogs' compares to their purebred progenitor breeds

Gina T. Bryson[1], Dan G. O'Neill[2], Claire L. Brand[1], Zoe Belshaw[3], Rowena M. A. Packer[1] *

1 Department of Clinical Science and Services, Royal Veterinary College, Hertfordshire, United Kingdom, 2 Department of Pathobiology and Population Sciences, The Royal Veterinary College, Hertfordshire, United Kingdom, 3 EviVet Evidence-Based Veterinary Consultancy, Nottingham, United Kingdom

* rpacker@rvc.ac.uk

**Data Availability Statement:** All relevant data are within the manuscript and its Supporting Information files.

## Abstract

Booming UK ownership of designer-crossbreed dogs resulting from intentional crossing of distinct purebred breeds is often motivated by perceived enhanced health, despite limited evidence supporting a strong 'hybrid vigour' effect in dogs. Improved evidence on the relative health of designer-crossbreed dogs could support prospective owners to make better acquisition decisions when choosing their new dog. This study used a cross-sectional survey of UK owners of three common designer-crossbreeds (Cavapoo, Cockapoo, and Labradoodle) and their progenitor breeds (Cavalier King Charles Spaniel, Cocker Spaniel, Labrador Retriever, and Poodle) to collect owner-reported health disorder information. The authors hypothesised that designer-crossbred breeds have lower odds of common disorders compared to their progenitor breeds. Multivariable analysis accounted for confounding between breeds: dog age, sex, neuter status, and owner age and gender. The odds for the 57 most common disorders were compared across the three designer-crossbreeds with each of their two progenitor breeds (342 comparisons). Valid responses were received for 9,402 dogs. The odds did not differ statistically significantly between the designer-crossbreeds and their relevant progenitor breeds in 86.6% (n = 296) of health comparisons. Designer-crossbreeds had higher odds for 7.0% (n = 24) of disorders studied, and lower odds for 6.4% (n = 22). These findings suggest limited differences in overall health status between the three designer-crossbreeds and their purebred progenitors, challenging widespread beliefs in positive hybrid vigour effects for health in this emerging designer-crossbreed demographic. Equally, the current study did not suggest that designer-crossbreeds have poorer health as has also been purported. Therefore, owners could more appropriately base acquisition decisions between designer-crossbreeds and their purebred progenitors on other factors important to canine welfare such as breeding conditions, temperament, conformation and health of parents.

**Funding:** R.M.A.P. (Kennel Club Charitable Trust - Research Grant) D.G.O. (Kennel Club Charitable Trust - International Canine Health Award) The funders had no role in study design, data collection and analysis, decision to publish, or preparation of the manuscript.

**Competing interests:** The authors have declared that no competing interests exist.

## Introduction

Over the past decade, as human motivations to own dogs based on their physical and heritage characteristics have continued to evolve, a new phenomenon called 'designer-crossbreeds' has emerged that is now a major contributor to the canine demographic landscape in many countries including the UK and USA [1–4]. Although almost all current pure breeds can be considered historically as crosses between other types of dogs (prior to their gene pools becoming closed in the registered pedigree populations) [5], the modern wave of designer-crossbreeding is different and is instead driven by the aim of deliberately creating 'hybrids' between existing pure breeds rather than to create a new breed *per se*. This designer crossbreeding phenomenon is widely considered to have been triggered in Australia in 1980 by Wally Conran, who then worked for the Royal Guide Dog Association. Conran intentionally crossed a Labrador Retriever with a Poodle with the aim of creating a non-shedding guide dog for a client whose husband was allergic to dog hair [6]. Faced with the challenge of finding homes for the remaining crossbred 'mongrel' puppies from the litter, Conran invoked some basic marketing principles to create public demand for his product by inventing a new, attractive name and a positive sales story that claimed hypoallergenicity and hybrid vigour in the absence of much supporting evidence [7, 8]. Hence the portmanteau name 'Labradoodle' was coined, with the resultant 'Labradoodle' puppies advertised as a new, and now desirable, hybrid breed with a supportive human and canine health-based backstory [6]. Demand for Labradoodles and subsequently several other 'designer' Poodle-crosses has grown rapidly over the intervening four decades [9]. The Cockapoo, a designer cross between a Cocker Spaniel and Poodle, was the second most popular puppy owned in the UK in 2019 [3] and demand for other Poodle crosses including the Cavapoo (designer cross between Cavalier King Charles Spaniel and Poodle) also notably increased in the UK during the COVID-19 pandemic in 2020 [10]. The progeny from this new wave of intentional crosses have been termed 'designer-crossbreeds' to distinguish them, and indeed increase the financial value of this category of dogs, from the traditional non-designer crossbreeds (also known as general crossbreeds, mixed breeds, or mongrels) and older crosses that are now considered as purebred dogs, e.g. the Silky Terrier (Fig 1).

As introduced above, using intentional crossbreeding between different defined types or breeds to create new types of dogs has been standard practice in dog breeding for over 150 years, with many of today's pure breed dogs resulting from various forms of designer-cross-breeding programmes during their original invention, though terms such as outcrossing may have been used back then to describe these breeding practices [11]. Therefore to avoid confusion, in the current paper, we will define pure breeds as types of dog where all the dogs within the breed have recognisably similar physical appearances and when bred together give rise to a subsequent generation that closely resemble the parental dogs, and therefore can be assumed to share high genetic similarity i.e., limited genetic diversity within the breed [12]. Within each pure breed, a pedigree subset of dogs is defined when a recognised breed register maintains a record of the stated forebearers for these dogs [13, 14]. For example, the Golden Retriever breed was recognised by The Kennel Club (UK) in 1913 after invention by crossbreeding between two pre-existing pure breeds: the Flat-coated Retriever and the now 'extinct' Tweed Water Spaniel [15]. In many ways, some modern designer-crossbreeds such as the Cockapoo and Labradoodle are now becoming so recognisable and saleable, with breed enthusiasts establishing their own canine registries, that they could already be considered to meet criteria as new, emerging pure breeds or even pedigree dogs [16, 17]. Yet despite dramatically rising demand to own designer-crossbreeds, anecdotal evidence suggests that a 'pure breed bias' form of discrimination still prevails, whereby the new wave of designer-crossbreeds are still tarred with a 'mongrel' moniker as a derogatory rather than simply a descriptive term [18].

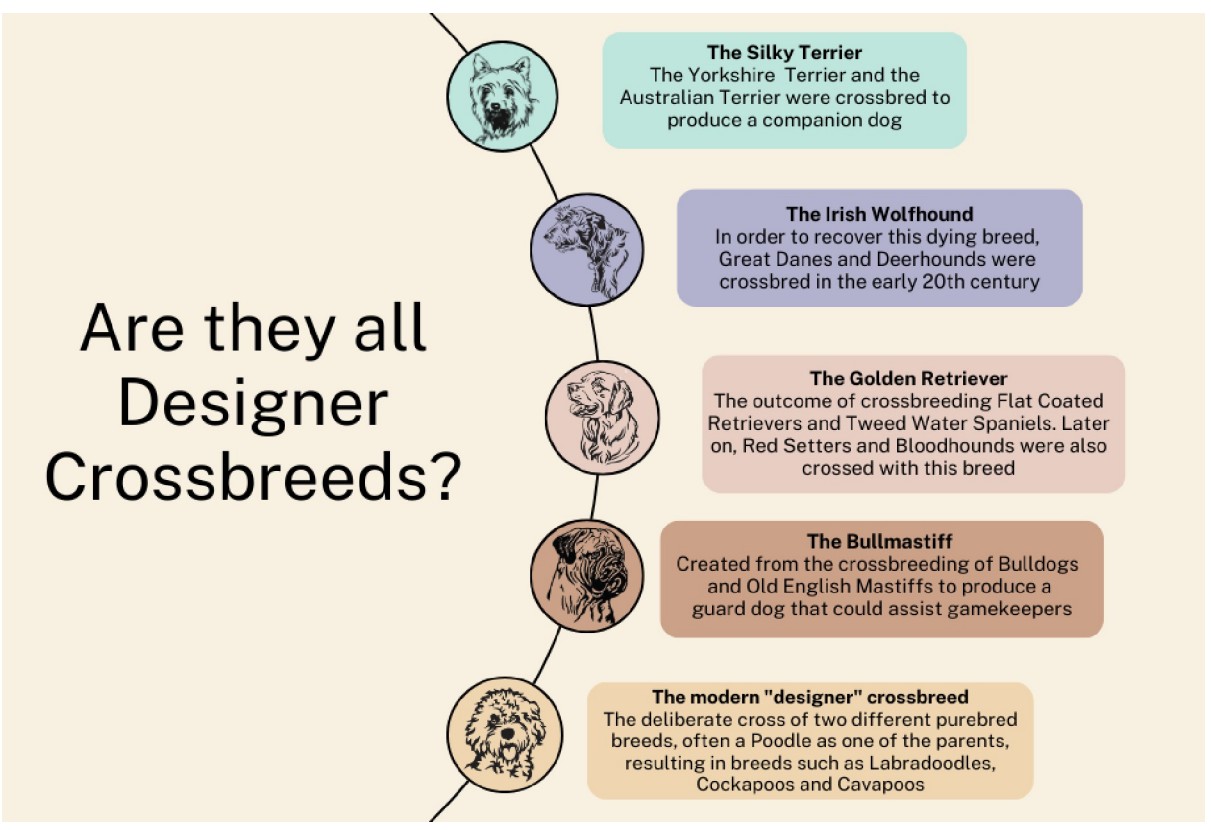

**Fig 1. A graphical representation illustrating how some current breeds were created from crossbreeding between pre-existing breeds.** Many of today's purebreds originated from crossbreeding between different pre-existing purebreds. The Silky Terrier, Golden Retriever and Bullmastiff were products of intentional breeding between different pre-existing breeds and therefore, could be considered 'designer dogs' that later gained purebred status. The Irish Wolfhound originated similarly but later on, and necessitated crossbreeding with other breeds like Great Danes in order to keep the Irish Wolfhound from disappearing altogether.

Modern designer-crossbreeds are often considered as somehow 'less' than the longer established pure breeds that also had designer-crossbred origins, such as the Golden Retriever [19–21], despite their similar origins differing only in the date when these breed types were originally invented. An improved evidence base on the relative health of the new wave of designer-crossbreeds compared to their progenitor pure bred parent breeds could contribute to a better understanding of these new designer types of dogs, and whether they have health-based deficiencies or advantages compared to their pure breed counterparts [22].

There is growing evidence of substantial predispositions to disorders in many current pure breeds of dogs resulting from genetic or conformational problems [17, 23–25]. Historically, skilled breeders were encouraged to improve the health and other aspects of their breed(s) by outcrossing (i.e., breeding between dogs of differing types or breeds) to augment health-related traits, for example, to increase genetic diversity or change conformational and/or behavioural traits [26]. However, since The Kennel Club's pedigree breed registers were formally closed to outcrossing in the UK in 1971, the opportunity for outcrossing to redress health, genetic and conformational problems within the pedigree subset of pure breeds has been largely removed, with just a few exceptions. Consequently, there is now growing evidence of certain serious genetic and conformational health issues that may require formally re-opening the stud books to outcrossing to resolve [12, 27]. In welcome moves to counter the issues resulting from closed studbooks, The Kennel Club is now increasingly open to registering dogs bred within

formally designed outcross programmes with pre-defined rationales necessitating the outcrossing. As an example, an outcross programme was created between Dalmatians and Pointers that aimed to reduce the prevalence of clinical problems associated with elevated levels of uric acid, a disorder inherited as an autosomal recessive fully penetrant gene and affecting one in four Dalmatians [28, 29]. Similarly, crossbreeding the Griffon Bruxellois with the Australian Terrier (to create hybrid 'Graussies' [30]) was undertaken to reduce the prevalence of the serious neurological disorder, Chiari malformation/syringomyelia (CM/SM) in the Griffon Bruxellois [30]. Sadly, however, despite the health success of these outcrossing programmes, there is still resistance from many 'traditional' breeders who hold that maintaining the purity of their breed by 'pure breeding', i.e., only selecting within their restricted gene pools, is a higher priority than protecting the health of future dogs of that breed using all genetic resources available within the wider dog population [26].

Outside the realms of pedigree dog registration systems, crossbreeding is still a mainstay breeding tool used by ethical breeders to protect canine health and improve other performance traits, e.g., behaviour. Crossbreeding is regularly used in the breeding of working dogs, such as Guide Dogs (commonly Golden Retriever x Labrador Retrievers), to retain favourable traits from one progenitor breed while displacing some of their health problems by incorporating genetic and conformational characteristics from an alternative progenitor breed [31]. When successful, this phenomenon is known as 'positive hybrid vigour', whereby the performance of the crossbred offspring for a particular trait is superior to the average of the purebred progenitor breeds [32]. Supporting at least some positive hybrid vigour effect in dogs, data on hip dysplasia between 1991–1995 in the US reported 10% lower prevalence in crossbreeds compared to purebreds [33]. Furthermore, purebred dogs were more likely to have two copies of the disease-associated allele (homozygous state) than crossbreeds, leading to the disease phenotype's expression [34]. However, the quality and strength of the evidence for an overall and meaningful positive hybrid vigour in crossbreed dogs has been challenged [32]. Inadequate specification of the progenitor breeds for the crossbreds under analysis has meant that most previous studies at best provides evidence for *apparent* hybrid vigour, as the crossbreeds may have derived from progenitors that already had higher/lower odds for that particular disorder. Furthermore, dogs that are 'designer-crossbreeds' (i.e., intentional crosses with known breed parentage) were included alongside non-designer crossbreeds/mixed breeds (with unknown heritage) in many studies as a generic 'crossbreed' grouping, obscuring effects of first-generation crosses vs. later genetic mixes [27, 35–37].

Belief in the existence of meaningful positive hybrid vigour effects in dogs has led to claims that crossbreed dogs are generally healthier, longer lived, and less prone to disease compared to their purebred progenitors [32, 38, 39]. However, despite intuitive allure, there is currently very limited data supporting the notion that a hybrid vigour effect results in substantially better overall health in novel designer-crossbreeds compared to their progenitors. Given this evidence vacuum, many owners appear to be using anecdotes and first principles to create belief systems about increased health in designer-crossbreeds that may be incorrect but yet has been documented as motivating their purchases and perpetuating popularity of these types of dogs [1]. An improved evidence base on the health of designer-crossbreeds relative to their progenitor breeds could assist prospective owners to use health-based rationale when making more informed decisions on which types of dogs to acquire. This is particularly important given that the concept of *negative* hybrid vigour has also been discussed, whereby crossbreeds exhibit worse health than their progenitors [32]. For example, analysis of anonymised primary care veterinary clinical data in the UK VetCompass research programme has identified that several designer-crossbreeds were at higher risk of otitis externa than their progenitor breeds, with Cavapoos, Cockapoos and Labradoodles among the ten breeds with the highest otitis externa

prevalence [40]. Consequently, a better evidence base is critical to ensure would-be owners are aware of both any health benefits or drawbacks of designer-crossbreeds prior to acquisition.

Therefore the current study aimed to address the current knowledge gap on the relative health of common designer-crossbreeds in the UK by exploring owner-reported health differences between designer-crossbreeds (Poodle crosses: Labradoodles, Cockapoos and Cavapoos) and their relevant purebred progenitor breeds. The study objectives were to estimate and compare the prevalence of common health disorders reported by UK owners between the three designer-crossbreeds and their relevant progenitor breeds aged up to five years using an online survey format. The study hypothesis was that in dogs aged up to five years, designer-crossbreeds have more reported disorders with reduced odds than increased odds compared to their progenitor breeds.

## Materials and methods

An online questionnaire was used to explore ownership experiences of selected designer-crossbreeds and their progenitor purebred breeds. The questionnaire was hosted using Research Electronic Data Capture (REDCap) software [41] and was open from 21st February 2023 to 21st April 2023. Participants typically took 20–25 minutes to complete the questionnaire. The study received ethical approval from the Social Science Research Ethical Review Board at the Royal Veterinary College (URN SR2022-0184). Participants gave their written consent via a tick box prior to starting the questionnaire. Participants could exit the survey at any time; however, because the survey was anonymous, respondents could not withdraw their response once submitted.

### Inclusion criteria

Participants were required to be aged 18 years or over, resident in the UK and to currently own at least one designer-crossbreed from a specified list of Poodle crosses [Cavapoo (CKCS crossed with a Poodle), Cockapoo (Cocker Spaniel crossed with a Poodle), Labradoodle (Labrador Retriever crossed with a Poodle)] or a purebred dog of a progenitor breed of the aforementioned designer-crossbreeds [CKCS, Cocker Spaniel, Labrador Retriever, Miniature Poodle, Standard Poodle or Toy Poodle]. The three designer-crossbreeds were chosen as the most numerous designer crosses in VetCompass [3].

Dogs were required to have been acquired aged 16 weeks or younger from 1st January 2019 onwards. This ensured all study dogs were aged under 5 years old at completion of the questionnaire to reduce age confounding effects on health [42, 43]. Participants with more than one eligible dog were asked to complete the questionnaire for their dog named first alphabetically.

### Survey content

The questionnaire comprised two parts.

1. The first section captured owner demographics, including age, gender, postcode and occupation, and canine demographics, including breed, sex and date of birth.

2. The second section captured the dog's health and diagnoses (if applicable) within the previous 12 months from the date of filling in the questionnaire, as well as any veterinary treatment received, with many of the questions adapted from previous 'Pandemic Puppy' surveys created by the same research group [4, 10]. Further questions explored specific health conditions with reported predispositions in the progenitor purebreds: Addison's disease, paroxysmal dyskinesia, epilepsy, hip and/or elbow dysplasia, Von Willebrand's

disease, cruciate ligament rupture, and cancers [44–47]. Most response options were presented as single-choice or multi-choice with an additional option for free-text responses. The questionnaire also included questions regarding the dog's behaviour and husbandry, that are not reported here. The full questionnaire is included in S1 Text.

## Survey recruitment

Participants were recruited by several pathways. Digital posters for individual designer-crossbreeds and purebred progenitor breeds (see S1 Fig) were shared by the authors via snowball sampling on relevant breed-specific pages on the social media sites (e.g., Facebook, Instagram, Reddit and X). Large UK animal charities including Blue Cross, RSPCA, and PDSA, veterinary-specific organisations (e.g., VetPartners Ltd), and other organisations from the sector (e.g., PetPlan, Pets at Home and APDAWG) promoted the questionnaire on their social media platforms. The pet classifieds site, Pets4Homes, shared the questionnaire link by direct email to users that had shown an interest in purchasing any of the study breeds/crossbreeds in the previous 600 days. The Kennel Club (UK) promoted the questionnaire via their registered breeders for the six purebred breeds and distributed in-person via breed-specific flyers at Crufts 2023, Birmingham. The complete list of disseminators is shown in the Acknowledgements section.

## Data processing

The questionnaire data were manually cleaned in Microsoft Excel (2013) to remove any responses missing the specified inclusion criteria, or ineligible responses. Free text disorder information was manually mapped to a VetCompass list of canine disorders using standardised VetCompass methodology [48].

## Data analysis

Statistical analysis used IBM SPSS Statistics (V29.0.0.0) software. Poodle responses were combined to represent all three Poodle-type breeds for analysis purposes. Descriptive statistics reported the frequency and percentage for relevant variables. Dog age (years) was calculated at the completion date of the questionnaire. Demographic variables were compared between designer-crossbreeds and purebreds using chi-square testing.

Multivariable binary logistic regression analysis evaluated each designer-crossbreed compared to their two progenitors as risk factors for each disorder. Each multivariable model included a fixed set of additional variables to account for confounding which were selected using an information theory approach [49] developed from other canine health studies that identified the following as confounding factors [50, 51]: dog age, sex, neuter status, insured status; and owner gender and age [43, 52]. Statistical significance was set at $p < 0.05$.

## Results

### Owner demographics

From a total of $n = 10,524$ responses received, $n = 1,113$ (10.6%) did not fully meet the inclusion criteria and $n = 9$ (0.1%) were ineligible responses. The final analytic dataset of 9,402 (89.3%) valid responses represented $n = 7,493$ female (79.7%) and $n = 1,012$ male respondents (10.8%) with other gender options for the remaining $n = 897$ (9.5%). Respondent gender distribution differed between designer-crossbreed and purebred owners, with female respondents

more likely to own designer-crossbreeds compared to purebreds (female respondents: designer-crossbreeds: 89.2% vs. purebred: 87.5%, $p = 0.020$). The most represented age group among all respondents was 45–54 years old (23.5%). The numbers in the age categories did not differ significantly between designer-crossbreed owners compared to purebred owners (designer-crossbreed owners: $n = 2,345$, 76% over 35; purebred owners: $n = 4,019$, 75.7% over 35; $p = 0.562$).

Overall, 5,132 (60.5%) respondents were first-time dog owners, while the remaining 3,353 (39.5%) respondents had previously owned a dog. Designer-crossbreed dogs were more likely to have first-time dog owners compared to purebred dogs (first-time owners: designer-cross-breeds: 49.6% vs. purebreds: 33.7%, $p<0.001$).

Most respondents lived in an adult-only home ($n = 4,864$, 51.73%). Designer-crossbreeds were more likely to live in households with children than purebred dogs (adult-only home: designer-crossbreeds: 53.0% vs. purebreds: 59.8%; $p<0.001$). Almost 10% of respondents ($n = 810$; 9.5%) worked in the canine/animal care sector. Owners of designer-crossbreed dogs were less likely to work in the canine/animal care sector compared to purebred owners (designer-crossbreeds: 4.9% vs purebreds: 12.1%; $p<0.001$).

## Dog demographics

The 9,402 dogs in the final analysis included 3,424 (36.4%) designer-crossbreed and 5,978 (63.6%) purebred dogs. The designer-crossbreed dogs included 985 (10.5% of 9,402 dogs) Cavapoo, 1,856 (19.7%) Cockapoo and 583 (6.2%) Labradoodle. The purebred dogs included 715 (7.6%) CKCS, 2,237 (23.8%) Cocker Spaniel, 2,099 (22.3%) Labrador Retriever, 352 (3.7%) Miniature Poodle, 315 (3.4%) Standard Poodle and 260 (2.8%) Toy Poodle ($n = 927$, 9.9% Poodles overall).

Among the $n = 3,424$ Poodle-cross designer-crossbreeds, Miniature Poodles ($n = 1,749$, 51.5%) was the most represented progenitor Poodle type in the sample. Of the three designer-crossbreed types, Cockapoo were the most likely to include a Miniature Poodle progenitor (Cockapoo: $n = 1,031$, 55.8%; Cavapoo: $n = 467$, 47.7%; Labradoodle: n = 251, 44.1%; $p<0.001$). Toy Poodle (884, 26.0%) was the second most represented progenitor Poodle type in the sample. Of the three designer-crossbreed types, Cavapoo were the most likely to include a Toy Poodle progenitor (Cavapoo: $n = 406$, 41.5%; Cockapoo: $n = 442$, 23.9%; Labradoodle: $n = 36$, 6.3%; $p<0.001$). Standard Poodle ($n = 426$, 12.50%) was the least commonly represented Poodle progenitor in the sample. Of the three designer-crossbreed types, Labradoodle were the most likely to include a Standard Poodle progenitor (Labradoodle: $n = 228$, 40.1%; Cockapoo: $n = 163$, 8.8%; Cavapoo: $n = 35$, 3.6%; $p<0.001$).

An 'F1' (first-generation cross between two purebred progenitor breeds) ($n = 1,899$, 20.2%) was the most commonly stated generational type overall across all three designer-crossbreeds. Labradoodles were significantly less likely to be an F1 cross compared to the other two designer-crossbreeds (F1: Cockapoo: $n = 1,103$, 59.70%; Cavapoo: $n = 600$, 61.3%; Labradoodle: $n = 196$, 33.8%; $p<0.001$).

The overall study population included $n = 4,373$ (50.5%) male and $n = 4,287$ (49.5%) female dogs. Designer-crossbreeds were more likely to be female than purebred dogs (designer-crossbreed: female $n = 1,638$, 51.3%, male: $n = 1,558$, 48.7%; purebred: female $n = 2,648$, 48.5%, male: $n = 2,815$, 51.5%; $p = 0.013$). The probability of being female did not differ between the three designer-crossbreeds (Cockapoo $n = 870$, 50.1%, Cavapoo $n = 429$, 46.9%, Labradoodle $n = 259$, 47.4%; $p = 0.237$). The overall study population included $n = 4,786$ (50.9%) entire dogs and $n = 3,869$ (41.1%) neutered dogs. The neuter status of n = 747 (8%) dogs was unspecified. Designer-crossbreeds were more likely to be neutered than purebreds (neutered:

designer-crossbreeds: $n$ = 1,650 (51.6%), purebreds: $n$ = 2,219 (40.6%); $p$<0.001). The probability of being neutered did not differ between the three designer-crossbreeds, (Cockapoo $n$ = 909, 52.4%, Cavapoo $n$ = 460, 50.2%, Labradoodle $n$ = 281, 51.6%; $p$ = 0.558).

Overall, $n$ = 7,432 (79.0%) of the study population were insured. Designer-crossbreeds were more likely to be insured than purebreds (insured: designer-crossbreeds $n$ = 2,826 (82.5%) purebreds $n$ = 4,606 (77.1%); $p$<0.001). The probability of being insured did not differ between the three designer-crossbreeds, (Cockapoo $n$ = 1,535, 88.4%, Cavapoo $n$ = 803, 87.6%, Labradoodle $n$ = 489, 89.6%; $p$ = 0.516).

## Disorder odds

Comparison of the odds between each of the three designer-crossbreeds and each of their two progenitor breeds in dogs aged up to five years across the 57 disorders (342 comparisons) did not identify statistical difference in 86.6% ($n$ = 296) disorder comparisons, with designer-crossbreeds having higher odds in 7.0% of disorders ($n$ = 24) and lower odds in 6.4% of disorders ($n$ = 22).

**Cockapoo vs. their two progenitor breeds.**   Compared to their Poodle progenitor breed, Cockapoos did not differ in their odds for 45 of 57 (79.0%) disorders after accounting for confounding in multivariable analyses (Table 1). Cockapoos had lower odds of five of 57 (9.0%) common disorders compared to Poodles: ophthalmological disorders (OR: 0.55, 95% CI: 0.42–0.72, $p$<0.001), patellar luxation (OR: 0.49, 95% CI: 0.29–0.81, $p$ = 0.005), weight loss (OR: 0.48, 95% CI: 0.23–0.99, $p$ = 0.046) food hypersensitivity/intolerance (OR: 0.42, 95% CI: 0.19–0.94, $p$ = 0.035) and dental disease (OR: 0.16, 95% CI: 0.03–0.87, $p$ = 0.033).

In contrast, Cockapoos had higher odds of seven of 57 (12.0%) disorders compared to Poodles: foreign bodies (OR: 3.53, 95% CI: 1.04–12.03, $p$ = 0.044), roundworm infestations (OR: 2.56, 95% CI: 1.42–4.62, $p$ = 0.002), anal sac disorders (OR: 2.09, 95% CI: 1.51–2.87, $p$<0.001), diarrhoea (OR: 1.63, 95% CI: 1.34–1.98, $p$<0.001), dietary indiscretion (OR: 1.59, 95% CI: 1.24–2.05, $p$<0.001), pruritus (OR: 1.50, 95% CI: 1.13–1.99, $p$ = 0.005) and vomiting (OR: 1.30, 95% CI: 1.06–1.59, $p$ = 0.011).

Compared to their Cocker Spaniel progenitor breed, Cockapoos had lower odds of five of 57 (4.0%) disorders: dietary indiscretion (OR:0.75, 95% CI: 0.63–0.89, $p$<0.001), lameness (OR:0.58, 95% CI: 0.41–0.84, $p$ = 0.004), multiple masses (OR: 0.56, 95% CI: 0.34–0.93, $p$ = 0.025), ophthalmological disorders (OR: 0.65, 95% CI: 0.52–0.81, $p$<0.001) and wounds (OR: 0.64, 95% CI: 0.48–0.86, $p$ = 0.003). Cockapoos had higher odds of three of 57 (5.0%) disorders studied: pruritus (OR:2.67, 95% CI:2.10–3.39, $p$<0.001), otitis externa (OR:2.13, 95% CI: 1.72–2.63, $p$<0.001) and vomiting (OR: 1.41, 95% CI: 1.21–1.65, $p$<0.001). The odds for the remaining 49 of 57 (86.0%) disorders did not differ statistically between the Cockapoo and Cocker Spaniels.

**Labradoodle vs. their two progenitor breeds.**   Compared to their Poodle progenitor breed, Labradoodles did not differ in their odds for 50 of 57 (88.0%) disorders after accounting for confounding in multivariable analyses.

Labradoodles had lower odds of one of 57 (2.0%) disorders studied compared to Poodles: patellar luxation (OR:0.24, 95% CI: 0.10–0.58, $p$ = 0.002) (Table 2). In contrast, Labradoodles had higher odds of six of 57 (11.0%) disorders compared to Poodles: allergies (OR:3.31, 95% CI: 1.36–14.16, $p$ = 0.013), alopecia (OR: 2.94, 95% CI: 0.87–9.96, $p$ = 0.083), dietary indiscretion (OR: 2.03, 95% CI: 1.48–2.76, $p$<0.001), wounds (OR: 1.82, 95% CI: 1.12–2.96, $p$ = 0.015), diarrhoea (OR:1.64, 95% CI: 1.27–2.11, $p$<0.001) and vomiting (OR:1.54, 95% CI: 1.20–1.99, $p$<0.001).

**Table 1. Descriptive and multivariable logistic regression analysis results comparing the probability of 57 common and/or important disorders between Cockapoo (n = 1856) dogs and their progenitor breeds, Poodle (n = 927) and Cocker Spaniel (n = 2237).**

| Disorder | Poodle (n = 927) | Cocker Spaniel (n = 2237) | Cockapoo (n = 1856) | Multivariable OR* Cockapoo vs Poodle | 95% CI** | P value | Multivariable OR* Cockapoo vs Cocker Spaniel | 95% CI** | P value |
|---|---|---|---|---|---|---|---|---|---|
| Addison's Disease | 16 (2.1%) | 13 (0.7%) | 15 (1.0%) | 0.70 | 0.32–1.53 | 0.376 | 1.42 | 0.65–3.11 | 0.378 |
| Adverse reaction to drug/vaccination | 4(0.5%) | 2 (0.1%) | 3 (0.2%) | 0.38 | 0.08–1.81 | 0.225 | 3.12 | 0.32–30.57 | 0.329 |
| Allergy/Allergic skin disorder | 4 (0.5%) | 16 (0.9%) | 14 (0.9%) | 1.53 | 0.49–4.76 | 0.462 | 1.12 | 0.52–2.39 | 0.778 |
| Alopecia | 4 (0.5%) | 16 (0.9%) | 9 (0.6%) | 0.97 | 0.29–3.21 | 0.954 | 0.59 | 0.25–1.36 | 0.216 |
| Anal sac disorder | 56 (7.3%) | 248 (13.6%) | 233 (15.1%) | 2.09 | 1.51–2.87 | **<0.001** | 1.08 | 0.88–1.33 | 0.487 |
| Anxious/distressed *** | 0 (0.0%) | 9 (0.5%) | 11 (0.7%) | ~ | ~ | ~ | ~ | ~ | ~ |
| Behaviour disorder | 2 (0.3%) | 7 (0.4%) | 3 (0.2%) | 0.48 | 0.07–3.55 | 0.475 | 0.41 | 0.08–2.03 | 0.275 |
| Cancer | 14 (1.8%) | 13 (0.7%) | 13 (0.8%) | 0.71 | 0.31–1.64 | 0.420 | 1.21 | 0.54–2.74 | 0.641 |
| Claw injury or Claw/nail disorder | 1 (0.1%) | 5 (0.3%) | 7 (0.5%) | 2.78 | 0.34–22.98 | 0.343 | 1.89 | 0.54–6.65 | 0.323 |
| Conjunctivitis | 1 (0.1%) | 3 (0.2%) | 1 (0.1%) | 0.51 | 0.03–8.62 | 0.642 | 0.47 | 0.05–4.78 | 0.519 |
| Coughing | 41 (5.3%) | 79 (4.3%) | 81 (5.3%) | 0.90 | 0.60–1.35 | 0.595 | 1.19 | 0.85–1.65 | 0.320 |
| Cruciate ligament rupture | 13 (1.7%) | 13 (0.7%) | 16 (1.0%) | 0.92 | 0.41–2.11 | 0.851 | 1.44 | 0.65–3.15 | 0.368 |
| Cryptorchidism | 4 (0.5%) | 8 (0.4%) | 4 (0.3%) | 0.38 | 0.09–1.58 | 0.183 | 0.46 | 0.14–1.57 | 0.214 |
| Dental disease | 6 (0.8%) | 3 (0.2%) | 2 (0.1%) | 0.16 | 0.03–0.87 | **0.033** | 0.75 | 0.12–4.54 | 0.751 |
| Dermatitis | 1 (0.1%) | 5 (0.3%) | 2 (0.1%) | 1.00 | 0.09–11.37 | 0.999 | 0.49 | 0.09–2.57 | 0.396 |
| Diarrhoea | 253 (33.0%) | 837 (45.7%) | 729 (47.4%) | 1.63 | 1.34–1.98 | **<0.001** | 1.08 | 0.93–1.25 | 0.310 |
| Dietary indiscretion | 101 (13.2%) | 439 (24.0%) | 320 (20.8%) | 1.59 | 1.24–2.05 | **<0.001** | 0.75 | 0.63–0.89 | **<0.001** |
| Dyspnoea | 6 (0.8%) | 4 (0.2%) | 5 (0.3%) | 0.41 | 0.12–1.41 | 0.157 | 1.61 | 0.42–6.18 | 0.488 |
| Elbow dysplasia/Elbow joint disorder *** | 0 (0.0%) | 1 (0.1%) | 0 (0.0%) | ~ | ~ | ~ | ~ | ~ | ~ |
| Epilepsy | 13 (1.7%) | 16 (0.9%) | 18 (1.2%) | 0.98 | 0.44–2.18 | 0.962 | 1.37 | 0.67–2.82 | 0.393 |
| Food hypersensitivity Food intolerance | 15 (2%) | 13 (0.7%) | 13 (0.8%) | 0.42 | 0.19–0.94 | **0.035** | 1.26 | 0.56–2.87 | 0.574 |
| Foreign body | 3 (0.4%) | 35 (1.9%) | 21 (1.4%) | 3.53 | 1.04–12.03 | **0.044** | 0.71 | 0.40–1.24 | 0.228 |
| Giardiasis | 3 (0.4%) | 2 (0.1%) | 6 (0.4%) | 0.71 | 0.15–3.33 | 0.668 | 4.77 | 0.53–43.33 | 0.165 |
| Grape/raisin intoxication | 2 (0.3%) | 7 (0.4%) | 5 (0.3%) | 1.97 | 0.23–17.03 | 0.538 | 0.75 | 0.23–2.40 | 0.623 |
| Heart murmur | 1 (0.1%) | 2 (0.1%) | 2 (0.1%) | 0.47 | 0.03–7.93 | 0.601 | 0.50 | 0.04–5.68 | 0.574 |
| Hernia *** | 0 (0.0%) | 3 (0.2%) | 2 (0.1%) | ~ | ~ | ~ | ~ | ~ | ~ |

*(Continued)*

**Table 1.** (Continued)

| Disorder | Poodle (n = 927) | Cocker Spaniel (n = 2237) | Cockapoo (n = 1856) | Multivariable OR* Cockapoo vs Poodle | 95% CI** | P value | Multivariable OR* Cockapoo vs Cocker Spaniel | 95% CI** | P value |
|---|---|---|---|---|---|---|---|---|---|
| Hip and/or elbow dysplasia | 16 (2.1%) | 32 (1.7%) | 27 (1.8%) | 1.04 | 0.53–2.05 | 0.913 | 1.01 | 0.59–1.75 | 0.962 |
| Hip dysplasia | 1 (0.1%) | 8 (0.4%) | 4 (0.3%) | 1.96 | 0.21–17.99 | 0.554 | 0.77 | 0.22–2.71 | 0.682 |
| Insect bite/sting | 2 (0.3%) | 11(0.6%) | 6 (0.4%) | 1.57 | 0.31–8.02 | 0.585 | 0.65 | 0.23–1.81 | 0.410 |
| Intoxication | 5 (0.7%) | 5 (0.3%) | 3 (0.2%) | 0.35 | 0.08–1.64 | 0.184 | 0.68 | 0.16–2.94 | 0.609 |
| Kennel Cough | 6 (0.8%) | 6 (0.3%) | 9 (0.6%) | 0.64 | 0.22–1.83 | 0.403 | 1.60 | 0.56–4.58 | 0.380 |
| Lameness | 35 (4.6%) | 100 (5.5%) | 52 (3.4%) | 0.65 | 0.41–1.03 | 0.066 | 0.58 | 0.41–0.84 | **0.004** |
| Limber tail *** | 0 (0.0%) | 1 (0.1%) | 0 (0.0%) | ~ | ~ | ~ | ~ | ~ | ~ |
| Multiple masses | 11 (1.4%) | 48 (2.6%) | 24 (1.6%) | 0.95 | 0.45–1.97 | 0.881 | 0.56 | 0.34–0.93 | **0.025** |
| Musculoskeletal injury *** | 0 (0.0%) | 6 (0.3%) | 8 (0.5%) | ~ | ~ | ~ | ~ | | ~ |
| Obesity | 6 (0.8%) | 41 (2.2%) | 29 (1.9%) | 1.95 | 0.78–4.91 | 0.155 | 0.67 | 0.39–1.14 | 0.139 |
| Ophthalmological disorders | 118 (15.4%) | 269 (14.7%) | 159 (10.3%) | 0.55 | 0.42–0.72 | **<0.001** | 0.65 | 0.52–0.81 | **<0.001** |
| Osteoarthritis | 2 (0.3%) | 7 (0.4%) | 2 (0.1%) | 0.78 | 0.07–8.75 | 0.837 | 0.32 | 0.07–1.58 | 0.162 |
| Otitis externa | 147 (19.2%) | 179 (9.8%) | 290 (18.8%) | 0.87 | 0.69–1.10 | 0.251 | 2.13 | 1.72–2.63 | **<0.001** |
| Overgrown nail(s) | 17 (2.2%) | 41 (2.2%) | 39 (2.5%) | 1.16 | 0.63–2.16 | 0.634 | 1.15 | 0.72–1.83 | 0.571 |
| Parasite infestation | 40 (5.2%) | 135 (7.4%) | 91 (5.9%) | 1.16 | 0.78–1.74 | 0.469 | 0.83 | 0.62–1.12 | 0.217 |
| Paroxysmal dyskinesia | 13 (1.7%) | 13 (0.7%) | 14 (0.9%) | 0.86 | 0.37–1.98 | 0.714 | 1.35 | 0.61–3.01 | 0.463 |
| Patellar luxation | 36 (4.7%) | 45 (2.4%) | 35 (2.3%) | 0.49 | 0.29–0.81 | **0.005** | 0.83 | 0.52–1.33 | 0.444 |
| Penile/prepuce disorder *** | 2 (0.3%) | 2 (0.1%) | 3 (0.2%) | ~ | ~ | ~ | ~ | ~ | ~ |
| Phantom pregnancy | 1 (0.1%) | 3 (0.2%) | 8 (0.5%) | 3.52 | 0.43–28.85 | 0.241 | 2.63 | 0.68–10.07 | 0.159 |
| Pruritus | 79 (10.3%) | 128 (7.0%) | 250 (16.2%) | 1.50 | 1.13–1.99 | **0.005** | 2.67 | 2.10–3.39 | **<0.001** |
| Roundworm infestation | 15 (2.0%) | 89 (4.9%) | 78 (5.1%) | 2.56 | 1.42–4.62 | **0.002** | 0.94 | 0.67–1.30 | 0.690 |
| Seizure disorder | 5 (0.7%) | 15 (0.8%) | 21 (1.4%) | 1.80 | 0.66–4.94 | 0.254 | 1.69 | 0.84–3.42 | 0.144 |
| Theobromine/chocolate intoxication | 6 (0.8%) | 8 (0.4%) | 5 (0.3%) | 0.40 | 0.11–1.42 | 0.157 | 0.74 | 0.23–2.39 | 0.616 |
| Traumatic injury *** | 0 (0.0%) | 7 (0.4%) | 4 (0.3%) | ~ | ~ | ~ | ~ | ~ | ~ |
| Umbilical hernia *** | 0 (0.0%) | 3 (0.2%) | 7 (0.5%) | ~ | ~ | ~ | ~ | ~ | ~ |
| Urinary incontinence | 8 (1.0%) | 19 (1.0%) | 17 (1.1%) | 1.32 | 0.51–3.45 | 0.567 | 0.99 | 0.50–1.97 | 0.973 |
| Urinary tract infection | 5 (0.7%) | 10 (0.5%) | 6 (0.4%) | 0.62 | 0.17–2.24 | 0.465 | 0.68 | 0.24–1.91 | 0.460 |

*(Continued)*

**Table 1.** (Continued)

| Disorder | Poodle (n = 927) | Cocker Spaniel (n = 2237) | Cockapoo (n = 1856) | Multivariable OR* Cockapoo vs Poodle | 95% CI** | P value | Multivariable OR* Cockapoo vs Cocker Spaniel | 95% CI** | P value |
|---|---|---|---|---|---|---|---|---|---|
| Vomiting | 233 (30.4%) | 575 (31.4%) | 594 (38.6%) | 1.30 | 1.06–1.59 | **0.011** | 1.41 | 1.21–1.65 | **<0.001** |
| Von Willebrand's Disease | 13 (1.7%) | 12 (0.7%) | 13 (0.8%) | 0.81 | 0.34–1.90 | 0.624 | 1.33 | 0.58–3.06 | 0.505 |
| Weight loss | 16 (2.1%) | 34 (1.9%) | 16 (1.0%) | 0.48 | 0.23–0.99 | **0.046** | 0.64 | 0.34–1.18 | 0.150 |
| Wound | 35 (4.6%) | 149 (8.1%) | 82 (5.3%) | 1.04 | 0.68–1.58 | 0.871 | 0.64 | 0.48–0.86 | **0.003** |

Multivariable modelling also included dog age, sex, neuter status, insured status, owner gender and owner age. Coloured cells denote the ranking of prevalence within the sample population (red is highest, yellow is middle and blue is lowest). If there are identical scores then the lower status colour is used. Prevalence in the sample is signified by the brackets.

*OR odds ratio.

**CI confidence interval.

*** indicate disorders where statistical analysis was not possible due to a disorder count of 0 (0.0%) from at least one purebred or designer-crossbreed. Bold text denotes a statistically significant result.

Compared to their Labrador Retriever progenitor breed, Labradoodles had lower odds of four of 57 (7.0%) disorders studied compared to Labrador Retrievers: hip and/or elbow dysplasia (OR:0.37, 95% CI: 0.17–0.81, $p = 0.014$), lameness (OR:0.35, 95% CI: 0.20–0.62, $p<0.001$), multiple masses (OR: 0.35, 95% CI: 0.20–1.72, $p = 0.029$) and wounds (OR: 0.52, 95% CI: 0.36–0.74, p<0.001). Labradoodles had higher odds of one of 57 (2.0%) disorders studied, otitis externa (OR: 1.80, 95% CI: 1.38–2.34, $p<0.001$).The odds for the remaining 52 of 57 (91.0%) disorders studied did not differ statistically between the Labradoodle and Labrador Retriever.

**Cavapoo vs. progenitor breeds.** Compared to their Poodle progenitor breed, Cavapoos did not differ in their odds for 50 of 57 (88.0%) disorders studied after accounting for confounding in multivariable analyses.

Cavapoos had lower odds of three of 57 (5.0%) disorders studied compared to Poodles: otitis externa (OR:0.68, 95% CI: 0.51–0.91, $p = 0.009$), ophthalmological disorders (OR:0.64, 95% CI: 0.47–0.87, $p = 0.005$) and lameness (OR:0.51, 95% CI: 0.28–0.94, $p = 0.032$) (Table 3). In contrast, Cavapoos had higher odds of four of 57 (7%) disorders studied compared to Poodles: anal sac disorders (OR:2.64, 95% CI: 1.85–3.75, $p<0.001$), diarrhoea (OR:1.55, 95% CI: 1.24–1.94, $p<0.001$), dietary indiscretion (OR:1.38, 95% CI: 1.03–1.84, $p = 0.031$) and vomiting (OR:1.27, 95% CI: 1.01–1.60, $p = 0.041$).

Compared to their CKCS progenitor breed, Cavapoos had lower odds for four of 57 (7.0%) disorders studied compared to CKCS: ophthalmological disorders (OR: 0.56, 95% CI: 0.40–0.77, $p<0.001$), anal sac disorders (OR: 0.53, 95% CI: 0.40–0.70, $p<0.001$), obesity (OR: 0.47, 95% CI: 0.24–0.94, $p = 0.032$) and overgrown nails (OR: 0.43, 95% CI: 0.20–0.93, $p = 0.033$). Cavapoos had higher odds for three of 57 (5.0%) disorders studied: vomiting (OR:1.92, 95% CI: 1.48–2.48, $p<0.001$), otitis externa (OR: 1.62, 95% CI: 0.51–0.91, $p = 0.009$) and diarrhoea (OR: 1.31, 95% CI: 1.03–1.66, $p = 0.031$). The odds for the remaining 50 of 57 (88.0%) disorders studied did not differ statistically between the Cavapoo and CKCS.

**Table 2. Descriptive and multivariable logistic regression analysis results comparing the probability of 57 common and/or important disorders between Labradoodle (n = 583) dogs and their progenitor breeds, Poodle (n = 927) and Labrador Retriever (n = 2099).**

| Disorders | Poodle (n = 927) | Labrador Retriever (n = 2099) | Labradoodle (n = 583) | Multivariable OR* Labradoodle vs Poodle | 95% CI** | P value | Multivariable OR * Labradoodle vs Labrador Retriever | 95% CI** | P value |
|---|---|---|---|---|---|---|---|---|---|
| Addison's Disease | 16 (2.1%) | 16 (0.9%) | 5 (1.0%) | 0.61 | 0.21–1.76 | 0.360 | 1.06 | 0.38–2.97 | 0.917 |
| Adverse reaction to drug/vaccination | 4 (0.5%) | 3 (0.2%) | 2 (0.4%) | 0.79 | 0.14–4.55 | 0.792 | 2.02 | 0.32–12.62 | 0.452 |
| Allergy/Allergic skin disorder | 4 (0.5%) | 32 (1.8%) | 11 (2.2%) | 4.39 | 1.36–14.16 | **0.013** | 1.33 | 0.65–2.72 | 0.443 |
| Alopecia | 4 (0.5%) | 60 (3.4%) | 8 (1.6%) | 2.94 | 0.87–9.96 | **0.083** | 0.57 | 0.27–1.23 | 0.150 |
| Anal sac disorder | 56 (7.3%) | 136 (7.7%) | 4 4(8.9%) | 1.15 | 0.75–1.78 | 0.518 | 1.23 | 0.85–1.79 | 0.281 |
| Anxious/distressed *** | 0 (0.0%) | 11 (0.6%) | 4 (0.8%) | ~ | ~ | ~ | ~ | ~ | ~ |
| Behaviour disorder | 2 (0.3%) | 6 (0.3%) | 2 (0.4%) | 1.43 | 0.19–10.60 | 0.726 | 1.13 | 0.22–5.85 | 0.887 |
| Cancer | 14 (1.8%) | 19 (1.1%) | 5 (1.0%) | 0.65 | 0.22–1.90 | 0.431 | 0.87 | 0.32–2.38 | 0.782 |
| Claw injury or Claw/nail disorder | 1 (0.1%) | 11 (0.6%) | 2 (0.4%) | 3.22 | 0.28–36.65 | 0.347 | 0.72 | 0.15–3.38 | 0.678 |
| Conjunctivitis | 1 (0.1%) | 8 (0.5%) | 1 (0.2%) | 1.78 | 0.11–29.17 | 0.686 | 0.43 | 0.05–3.53 | 0.429 |
| Coughing | 41 (5.3%) | 103 (5.9%) | 37 (7.5%) | 1.27 | 0.78–2.08 | 0.337 | 1.17 | 0.78–1.77 | 0.454 |
| Cruciate ligament rupture | 13 (1.7%) | 19 (1.1%) | 5 (1.0%) | 0.73 | 0.25–2.18 | 0.574 | 0.83 | 0.30–2.28 | 0.714 |
| Cryptorchidism | 4 (0.5%) | 3 (0.2%) | 1 (0.2%) | 0.34 | 0.04–3.35 | 0.356 | 0.94 | 0.09–9.88 | 0.962 |
| Dental disease | 6 (0.8%) | 6 (0.3%) | 1 (0.2%) | 0.29 | 0.03–2.53 | 0.260 | 0.53 | 0.06–4.55 | 0.563 |
| Dermatitis *** | 1 (0.1%) | 0 (0.0%) | 0 (0.0%) | ~ | ~ | ~ | ~ | ~ | ~ |
| Diarrhoea | 253 (33.0%) | 771 (43.8%) | 228 (46.3%) | 1.64 | 1.27–2.11 | **<0.001** | 1.07 | 0.86–1.33 | 0.574 |
| Dietary indiscretion | 101 (13.2%) | 392 (22.3%) | 123 (25.0%) | 2.03 | 1.48–2.76 | **<0.001** | 0.98 | 0.76–1.26 | 0.867 |
| Dyspnoea | 6 (0.8%) | 6 (0.3%) | 1 (0.2%) | 0.22 | 0.03–1.87 | 0.164 | 0.52 | 0.06–4.49 | 0.555 |
| Elbow dysplasia/Elbow joint disorder *** | 0 (0.0%) | 11 (0.6%) | 0 (0.0%) | ~ | ~ | ~ | ~ | ~ | ~ |
| Epilepsy | 13 (1.7%) | 16 (0.9%) | 7 (1.4%) | 1.05 | 0.39–2.81 | 0.927 | 1.50 | 0.60–3.77 | 0.385 |
| Food hypersensitivity Food intolerance | 15 (2.0%) | 16 (0.9%) | 5 (1.0%) | 0.74 | 0.25–2.15 | 0.574 | 1.60 | 0.55–4.63 | 0.384 |
| Foreign body | 3 (0.4%) | 23 (1.3%) | 5 (1.0%) | 3.29 | 0.76–14.23 | 0.111 | 0.96 | 0.35–2.61 | 0.933 |
| Giardiasis | 3 (0.4%) | 5 (0.3%) | 2 (0.4%) | 1.26 | 0.20–7.98 | 0.808 | 1.05 | 0.19–5.65 | 0.959 |
| Grape/raisin intoxication | 2 (0.3%) | 10 (0.6%) | 4 (0.8%) | 6.23 | 0.68–57.34 | 0.106 | 1.37 | 0.41–4.53 | 0.610 |
| Heart murmur *** | 1 (0.1%) | 2 (0.1%) | 0 (0.0%) | ~ | ~ | ~ | ~ | ~ | ~ |
| Hernia *** | 0 (0.0%) | 1 (0.1%) | 1 (0.2%) | ~ | ~ | ~ | ~ | ~ | ~ |
| Hip and/or elbow dysplasia | 16 (2.1%) | 69 (3.9%) | 7 (1.4%) | 0.72 | 0.29–1.82 | 0.489 | 0.37 | 0.17–0.81 | **0.014** |

*(Continued)*

**Table 2.** (Continued)

| Disorders | Poodle (n = 927) | Labrador Retriever (n = 2099) | Labradoodle (n = 583) | Multivariable OR* Labradoodle vs Poodle | 95% CI** | P value | Multivariable OR * Labradoodle vs Labrador Retriever | 95% CI** | P value |
|---|---|---|---|---|---|---|---|---|---|
| Hip dysplasia *** | 1 (0.1%) | 15 (0.9%) | 0 (0.0%) | ~ | ~ | ~ | ~ | ~ | ~ |
| Insect bite/sting | 2 (0.3%) | 4 (0.2%) | 2 (0.4%) | 1.69 | 0.22–12.80 | 0.613 | 2.35 | 0.37–14.99 | 0.365 |
| Intoxication *** | 5 (0.7%) | 15 (0.9%) | 0 (0.0%) | ~ | ~ | ~ | ~ | ~ | ~ |
| Kennel Cough | 6 (0.8%) | 11 (0.6%) | 2 (0.4%) | 0.43 | 0.09–2.21 | 0.314 | 0.58 | 0.13–2.70 | 0.490 |
| Lameness | 35 (4.6%) | 149 (8.5%) | 14 (2.8%) | 0.58 | 0.31–1.10 | 0.097 | 0.35 | 0.20–0.62 | **<0.001** |
| Limber tail *** | 0 (0.0%) | 14 (0.8%) | 1 (0.2%) | ~ | ~ | ~ | ~ | ~ | ~ |
| Multiple masses | 11 (1.4%) | 53 (3.0%) | 5 (1.0%) | 0.59 | 0.20–1.72 | 0.332 | 0.35 | 0.14–0.90 | **0.029** |
| Musculoskeletal injury *** | 0 (0.0%) | 6 (0.3%) | 1 (0.2%) | ~ | ~ | ~ | ~ | ~ | ~ |
| Obesity | 6 (0.8%) | 42 (2.4%) | 10 (2.0%) | 2.56 | 0.90–7.30 | 0.079 | 0.91 | 0.44–1.86 | 0.791 |
| Ophthalmological disorders | 118 (15.4%) | 270 (15.3%) | 77 (15.7%) | 0.93 | 0.67–1.29 | 0.648 | 0.99 | 0.74–1.32 | 0.919 |
| Osteoarthritis *** | 2 (0.3%) | 21 (1.2%) | 0 (0.0%) | ~ | ~ | ~ | ~ | ~ | ~ |
| Otitis externa | 147 (19.2%) | 266 (15.1%) | 116 (23.6%) | 1.18 | 0.88–1.59 | 0.260 | 1.80 | 1.38–2.34 | **<0.001** |
| Overgrown nail(s) | 17 (2.2%) | 60 (3.4%) | 15 (3.0%) | 1.11 | 0.51–2.43 | 0.795 | 0.74 | 0.39–1.41 | 0.362 |
| Parasite infestation | 40 (5.2%) | 72 (4.1%) | 23 (4.7%) | 0.80 | 0.45–1.41 | 0.432 | 1.08 | 0.64–1.82 | 0.768 |
| Paroxysmal dyskinesia | 13 (1.7%) | 13 (0.7%) | 5 (1.0%) | 0.73 | 0.24–2.19 | 0.577 | 1.18 | 0.41–3.41 | 0.757 |
| Patellar luxation | 36 (4.7%) | 23 (1.3%) | 6 (1.2%) | 0.24 | 0.10–0.58 | **0.002** | 0.81 | 0.32–2.02 | 0.648 |
| Penile/prepuce disorder *** | 2 (0.3%) | 1 (0.1%) | 0 (0.0%) | ~ | ~ | ~ | ~ | ~ | ~ |
| Phantom pregnancy *** | 1 (0.1%) | 10 (0.6%) | 0 (0.0%) | ~ | ~ | ~ | ~ | ~ | ~ |
| Pruritus | 79 (10.3%) | 209 (11.9%) | 60 (12.2%) | 1.13 | 0.78–1.65 | 0.510 | 1.14 | 0.83–1.58 | 0.414 |
| Roundworm infestation | 15 (2.0%) | 50 (2.8%) | 13 (2.6%) | 1.38 | 0.63–3.02 | 0.419 | 0.77 | 0.41–1.45 | 0.419 |
| Seizure disorder | 5 (0.7%) | 30 (1.7%) | 7 (1.4%) | 2.45 | 0.75–8.00 | 0.137 | 1.04 | 0.44–2.44 | 0.938 |
| Theobromine/chocolate intoxication | 6 (0.8%) | 5 (0.3%) | 1 (0.2%) | 0.28 | 0.03–2.48 | 0.253 | 0.67 | 0.08–5.86 | 0.713 |
| Traumatic injury *** | 0 (0.0%) | 6 (0.3%) | 3 (0.6%) | ~ | ~ | ~ | ~ | ~ | ~ |
| Umbilical hernia *** | 0 (0.0%) | 1 (0.1%) | 3 (0.6%) | ~ | ~ | ~ | ~ | ~ | ~ |
| Urinary incontinence | 8 (1.0%) | 16 (0.9%) | 6 (1.2%) | 1.38 | 0.43–4.43 | 0.593 | 1.32 | 0.50–3.46 | 0.575 |
| Urinary tract infection | 5 (0.7%) | 8 (0.5%) | 7 (1.4%) | 1.87 | 0.51–6.87 | 0.344 | 2.28 | 0.77–6.78 | 0.137 |
| Vomiting | 233 (30.4%) | 646 (36.7%) | 200 (40.7%) | 1.54 | 1.20–1.99 | **<0.001** | 1.22 | 0.98–1.52 | 0.079 |
| Von Willebrand's Disease | 13 (1.7%) | 12 (0.7%) | 1 (0.2%) | 0.72 | 0.24–2.17 | 0.562 | 1.25 | 0.43–3.66 | 0.681 |
| Weight loss | 16 (2.1%) | 37 (2.1%) | 6 (1.2%) | 0.66 | 0.25–1.74 | 0.397 | 0.77 | 0.32–1.89 | 0.574 |

*(Continued)*

**Table 2.** (Continued)

| Disorders | Poodle (n = 927) | Labrador Retriever (n = 2099) | Labradoodle (n = 583) | Multivariable OR* Labradoodle vs Poodle | 95% CI** | P value | Multivariable OR * Labradoodle vs Labrador Retriever | 95% CI** | P value |
|---|---|---|---|---|---|---|---|---|---|
| Wound | 35 (4.6%) | 286 (16.3%) | 40 (8.1%) | 1.82 | 1.12–2.96 | **0.015** | 0.52 | 0.36–0.74 | **<0.001** |

Multivariable modelling also included dog age, sex, neuter status, insured status, owner gender and owner age. Coloured cells denote the ranking of prevalence within the sample population (red is highest, yellow is middle and blue is lowest). If there are identical scores then the lower status colour is used. Prevalence in the sample is signified by the brackets.

*OR odds ratio.

**CI confidence interval.

*** indicate disorders where statistical analysis was not possible due to a disorder count 0 (0.0%) from at least one purebred or designer-crossbreed. Bold text denotes a statistically significant result.

## Discussion

This study used a questionnaire to capture and explore ownership experiences regarding designer-crossbreeds and their progenitor breeds. The study focused on the health of three popular UK designer-crossbreeds that share Poodle as a common progenitor, and compared this to their progenitor purebred breeds in the UK. Comparison of the odds between each of three designer-crossbreeds and each of their two progenitor breeds in dogs aged up to five years across the 57 common disorders (342 comparisons) did not identify a statistical difference in 86.6% (n = 296) comparisons, with designer-crossbreeds having higher odds in 7.0% (n = 24) and lower odds in 6.4% (n = 22). Overall, these findings provide no evidence for a meaningful difference in overall health between these three designer-crossbreeds and their relevant progenitor breeds. The results therefore offer little support for the study hypothesis, and widespread common belief, of enhanced overall health profiles in designer-crossbreeds as a result of positive hybrid vigour. Instead, the results suggest that the overall health of this emerging designer-crossbreed demographic is largely similar to their progenitor breeds. This is particularly pertinent when referring back to Fig 1 which illustrates how many of today's purebreds originated from crossbreeding between distinct purebred breeds and were therefore designer-crossbreeds themselves during the early phase of their purebred development.

These findings conversely challenge the 'pure bred bias' that maintains that indiscriminate breeding outside of purebred lines will automatically lead to crossbreed dogs that have poorer health than their purebred parent breeds [21, 53, 54].

The current study's results differ from some of the pre-existing, albeit limited, health data available which had suggested a substantial health advantage in some specific designer-crossbreeds compared to purebreds. For example, a US pet insurance provider analysed cancer diagnosis claims for 1.61 million dogs over a six-year period, reporting that owners of Labradoodle and Goldendoodle were significantly less likely to submit a claim related to cancer diagnoses/treatment compared to their progenitor breeds [55]. Given the current study only explored health in dogs aged under five, it is possible that later differences in the risk of disorders such as cancers that are associated with more aged dogs [56] could emerge as these populations age. In the current study, reduced disorder prevalence in designer-crossbreeds was identified for a limited number of disorders, and an almost equal proportion of disorders were found to be of higher prevalence in this population. This novel finding supports an overall health equivalence between these three designer-crossbreeds and their relevant progenitors, at least while under five years old, and is particularly pertinent given public perceptions of better

**Table 3. Descriptive and multivariable logistic regression analysis results comparing the probability of 57 common and/or important disorders between Cavapoo (n = 985) dogs and their progenitor breeds, Poodle (n = 927) and CKCS (n = 715).**

| Disorders | Poodle (n = 927) | CKCS (n = 715) | Cavapoo (n = 985) | Multivariable OR* Cavapoo vs Poodle | 95% CI** | P value | Multivariable OR* Cavapoo vs CKCS | 95% CI** | P value |
|---|---|---|---|---|---|---|---|---|---|
| Addison's Disease | 16 (2.1%) | 4(0.7%) | 7(0.9%) | 0.50 | 0.18–1.37 | 0.176 | 1.19 | 0.32–4.39 | 0.794 |
| Adverse reaction to drug/vaccination | 4 (0.5%) | 3(0.5%) | 3(0.4%) | 0.71 | 0.15–3.34 | 0.668 | 0.61 | 0.12–3.13 | 0.555 |
| Allergy/Allergic skin disorder | 4 (0.5%) | 3(0.5%) | 4(0.5%) | 1.31 | 0.31–5.65 | 0.714 | 1.38 | 0.29–6.59 | 0.690 |
| Alopecia | 4 (0.5%) | 9(1.6%) | 5(0.6%) | 1.37 | 0.35–5.44 | 0.654 | 0.60 | 0.18–1.95 | 0.392 |
| Anal sac disorder | 56 (7.3%) | 168 (30.1%) | 131 (16.6%) | 2.64 | 1.85–3.75 | **<0.001** | 0.53 | 0.40–0.70 | **<0.001** |
| Anxious/distressed *** | 0 (0.0%) | 0(0.0%) | 3(0.4%) | ~ | ~ | ~ | ~ | ~ | ~ |
| Behaviour disorder | 2 (0.3%) | 2(0.4%) | 5(0.6%) | 1.14 | 0.18–7.13 | 0.893 | 1.93 | 0.19–19.38 | 0.577 |
| Cancer | 14 (1.8%) | 4(0.7%) | 6(0.8%) | 0.47 | 0.16–1.40 | 0.174 | 1.12 | 0.26–4.88 | 0.879 |
| Claw injury or Claw/nail disorder *** | 1 (0.1%) | 0(0.0%) | 3(0.4%) | ~ | ~ | ~ | ~ | ~ | ~ |
| Conjunctivitis *** | 1 (0.1%) | 0(0.0%) | 1(0.1%) | ~ | ~ | ~ | ~ | ~ | ~ |
| Coughing | 41 (5.3%) | 36(6.5%) | 41(5.2%) | 0.85 | 0.53–1.37 | 0.501 | 0.82 | 0.50–1.35 | 0.439 |
| Cruciate ligament rupture | 13 (1.7%) | 2(0.4%) | 6(0.8%) | 0.53 | 0.17–1.61 | 0.260 | 1.66 | 0.31–8.89 | 0.556 |
| Cryptorchidism | 4 (0.5%) | 3(0.5%) | 1(0.1%) | 0.20 | 0.02–1.82 | 0.153 | 0.22 | 0.02–2.17 | 0.194 |
| Dental disease | 6 (0.8%) | 1(0.2%) | 1(0.1%) | 0.15 | 0.02–1.37 | 0.093 | 0.46 | 0.03–7.67 | 0.590 |
| Dermatitis | 1 (0.1%) | 2(0.4%) | 2(0.3%) | 2.13 | 0.17–26.63 | 0.557 | 0.83 | 0.10–6.70 | 0.862 |
| Diarrhoea | 253 (33%) | 223 (40.0%) | 355 (44.9%) | 1.55 | 1.24–1.94 | **<0.001** | 1.31 | 1.03–1.66 | **0.031** |
| Dietary indiscretion | 101 (13.2%) | 92 (16.5%) | 154 (19.5%) | 1.38 | 1.03–1.84 | **0.031** | 0.97 | 0.72–1.32 | 0.864 |
| Dyspnoea *** | 6 (0.8%) | 12(2.2%) | 0(0.0%) | ~ | ~ | ~ | ~ | ~ | ~ |
| Elbow dysplasia/Elbow joint disorder *** | 0 (0.0%) | 2(0.4%) | 0(0.0%) | ~ | ~ | ~ | ~ | ~ | ~ |
| Epilepsy | 13 (1.7%) | 3(0.5%) | 6(0.8%) | 0.5 | 0.16–1.52 | 0.222 | 1.06 | 0.24–4.65 | 0.934 |
| Food hypersensitivity Food intolerance | 15 (2.0%) | 4(0.7%) | 5(0.6%) | 0.40 | 0.14–1.15 | 0.090 | 1.03 | 0.27–3.99 | 0.966 |
| Foreign body | 3 (0.4%) | 10(1.8%) | 7(0.9%) | 2.42 | 0.56–10.58 | 0.239 | 0.48 | 0.16–1.43 | 0.188 |
| Giardiasis | 3 (0.4%) | 3(0.5%) | 4 (0.5%) | 0.91 | 0.18–4.51 | 0.904 | 0.58 | 0.12–2.88 | 0.501 |
| Grape/raisin intoxication | 2 (0.3%) | 2(0.4%) | 1 (0.1%) | 1.18 | 0.07–21.09 | 0.910 | 0.44 | 0.04–5.35 | 0.518 |
| Heart murmur | 1 (0.1%) | 8(1.4%) | 3(0.4%) | 3.15 | 0.31–32.03 | 0.333 | 0.49 | 0.12–2.04 | 0.326 |
| Hernia *** | 0(0.0%) | 6(1.1%) | 1(0.1%) | ~ | ~ | ~ | ~ | ~ | ~ |
| Hip and/or elbow dysplasia | 16 (2.1%) | 11(2.0%) | 9(1.1%) | 0.58 | 0.23–1.44 | 0.240 | 0.56 | 0.21–1.47 | 0.237 |

*(Continued)*

**Table 3.** (Continued)

| Disorders | Poodle (n = 927) | CKCS (n = 715) | Cavapoo (n = 985) | Multivariable OR* Cavapoo vs Poodle | 95% CI** | P value | Multivariable OR* Cavapoo vs CKCS | 95% CI** | P value |
|---|---|---|---|---|---|---|---|---|---|
| Hip dysplasia | 1 (0.1%) | 4(0.7%) | 1(0.1%) | 0.86 | 0.05–14.55 | 0.914 | 0.18 | 0.02–1.70 | 0.133 |
| Insect bite/sting | 2 (0.3%) | 1(0.2%) | 2(0.3%) | 1.12 | 0.15–8.39 | 0.916 | 1.68 | 0.14–19.70 | 0.681 |
| Intoxication *** | 5 (0.7%) | 1(0.2%) | 0(0.0%) | ~ | ~ | ~ | ~ | ~ | ~ |
| Kennel Cough | 6 (0.8%) | 2(0.4%) | 2(0.3%) | 0.24 | 0.05–1.24 | 0.089 | 0.72 | 0.10–5.27 | 0.744 |
| Lameness | 35 (4.6%) | 25(4.5%) | 17(2.1%) | 0.51 | 0.28–0.94 | **0.032** | 0.57 | 0.30–1.09 | 0.088 |
| Limber tail *** | 0 (0.0%) | 0(0.0%) | 0(0.0%) | ~ | ~ | ~ | ~ | ~ | ~ |
| Multiple masses | 11 (1.4%) | 16(2.9%) | 16(2.0%) | 1.53 | 0.67–3.50 | 0.311 | 1.16 | 0.53–2.55 | 0.705 |
| Musculoskeletal injury *** | 0 (0.0%) | 4(0.7%) | 3(0.4%) | ~ | ~ | ~ | ~ | ~ | ~ |
| Obesity | 6 (0.8%) | 27(4.8%) | 15(1.9%) | 2.18 | 0.81–5.87 | 0.123 | 0.47 | 0.24–0.94 | **0.032** |
| Ophthalmological disorders | 118 (15.4%) | 106 (19.0%) | 90 (11.4%) | 0.64 | 0.85–0.87 | **0.005** | 0.56 | 0.40–0.77 | **<0.001** |
| Osteoarthritis | 2 (0.3%) | 4(0.7%) | 2(0.3%) | 1.75 | 0.15–20.00 | 0.655 | 0.42 | 0.07–2.43 | 0.332 |
| Otitis externa | 147 (19.2%) | 59 (10.6%) | 115 (14.5%) | 0.68 | 0.51–0.91 | **0.009** | 1.62 | 1.13–2.31 | **0.009** |
| Overgrown nail(s) | 17 (2.2%) | 21(3.8%) | 11(1.4%) | 0.75 | 0.33–1.70 | 0.495 | 0.43 | 0.20–0.93 | **0.033** |
| Parasite infestation | 40 (5.2%) | 51(9.1%) | 45(5.7%) | 1.03 | 0.64–1.66 | 0.901 | 0.63 | 0.40–1.00 | 0.051 |
| Paroxysmal dyskinesia | 13 (1.7%) | 2(0.4%) | 6(0.8%) | 0.53 | 0.17–1.61 | 0.260 | 1.66 | 0.31–8.89 | 0.556 |
| Patellar luxation | 36 (4.7%) | 30(5.4%) | 27(3.4%) | 0.78 | 0.45–1.36 | 0.385 | 0.71 | 0.40–1.26 | 0.241 |
| Penile/prepuce disorder *** | 2 (0.3%) | 0(0.0%) | 2(0.3%) | ~ | ~ | ~ | ~ | ~ | ~ |
| Phantom pregnancy | 1 (0.1%) | 1(0.2%) | 2(0.3%) | 2.73 | 0.22–33.81 | 0.434 | 1.18 | 0.10–14.24 | 0.899 |
| Pruritus | 79 (10.3%) | 68 (12.2%) | 96 (12.1%) | 1.16 | 0.83–1.62 | 0.399 | 1.20 | 0.84–1.71 | 0.322 |
| Roundworm infestation | 15 (2.0%) | 14(2.5%) | 17(2.1%) | 0.95 | 0.45–1.99 | 0.888 | 0.72 | 0.34–1.50 | 0.375 |
| Seizure disorder | 5 (0.7%) | 10(1.8%) | 5(0.6%) | 0.83 | 0.23–3.00 | 0.777 | 0.37 | 0.12–1.15 | 0.084 |
| Theobromine/chocolate intoxication | 6 (0.8%) | 1(0.2%) | 1(0.1%) | 0.22 | 0.02–2.03 | 0.182 | 0.68 | 0.04–11.66 | 0.792 |
| Traumatic injury *** | 0 (0.0%) | 0(0.0%) | 1(0.1%) | ~ | ~ | ~ | ~ | ~ | ~ |
| Umbilical hernia *** | 0 (0.0%) | 3(0.5%) | 3(0.4%) | ~ | ~ | ~ | ~ | ~ | ~ |
| Urinary incontinence | 8 (1.0%) | 6(1.1%) | 3(0.4%) | 0.59 | 0.14–2.49 | 0.473 | 0.34 | 0.08–1.42 | 0.139 |
| Urinary tract infection | 5 (0.7%) | 2(0.4%) | 1(0.1%) | 0.21 | 0.02–1.98 | 0.174 | 0.34 | 0.03–3.93 | 0.387 |
| Vomiting | 233 (30.4%) | 142 (25.4%) | 294 (37.2%) | 1.27 | 1.01–1.60 | **0.041** | 1.92 | 1.48–2.48 | **<0.001** |
| Von Willebrand's Disease | 13 (1.7%) | 2(0.4%) | 6(0.8%) | 0.53 | 0.17–1.61 | 0.260 | 1.66 | 0.31–8.89 | 0.556 |
| Weight loss | 16 (2.1%) | 11 (2.0%) | 12 (1.5%) | 0.76 | 0.33–1.71 | 0.503 | 0.82 | 0.34–1.98 | 0.664 |

*(Continued)*

**Table 3.** (Continued)

| Disorders | Poodle (n = 927) | CKCS (n = 715) | Cavapoo (n = 985) | Multivariable OR* Cavapoo vs Poodle | 95% CI** | P value | Multivariable OR* Cavapoo vs CKCS | 95% CI** | P value |
|---|---|---|---|---|---|---|---|---|---|
| Wound | 35 (4.6%) | 25 (4.5%) | 30 (3.8%) | 0.76 | 0.45–1.29 | 0.307 | 1.07 | 0.60–1.90 | 0.830 |

Multivariable modelling also included dog age, sex, neuter status, insured status, owner gender and owner age. Coloured cells denote the ranking of prevalence within the sample population (red is highest, yellow is middle and blue is lowest). If there are identical scores then the lower status colour is used. Prevalence in the sample is signified by the brackets.

*OR odds ratio.

**CI confidence interval.

*** indicate disorders where statistical analysis was not possible due to a disorder count of 0 (0.0%) from at least one purebred or designer-crossbreed. Bold text denotes a statistically significant result.

general health compared to purebreds is a major purchasing motivator for many owners of designer-crossbreed dogs [1].

Although results from the current study predominantly suggest minimal overall health differences between designer-crossbreeds and their progenitor breeds, several notable health trends were identified. Having accounted for confounding factors, all three designer-crossbreeds in the current study had higher odds of dietary indiscretion, vomiting and diarrhoea than the Poodle that was their shared progenitor breed. Labradoodle and Cockapoo did not differ from their non-Poodle progenitors regarding these specific disorders, but Cavapoo also had higher odds for vomiting than the CKCS. Numerous studies exploring health and longevity in UK dogs reveal enteropathy as a highly prevalent canine disorder in the UK [52, 57, 58]. Higher odds of enteropathy in designer-crossbreeds than Poodles could reflect the influence of their non-Poodle progenitor; for example, Labrador Retrievers were previously reported with higher odds compared to non-Labrador Retrievers for enteropathy [59, 60], which may have contributed to the increased odds observed in Labradoodles compared to Poodles. Similarly, enteropathy was the sixth most prevalent disorder in Cocker Spaniels [61] which may have contributed to the increased odds observed in the current study in Cockapoo compared to Poodles. The current study's findings may help veterinary professionals tailor diagnostics, treatments and/or advice for owners of these specific designer-crossbreeds, including dietary modification and preventative care, although further study into specific disorder prevalence and reasons contributing to this are necessary to support this study's results.

There was a trend towards higher odds of dermatological disorders in designer-crossbreeds compared to their progenitors in the current study. Cockapoo had higher odds for pruritus than both progenitor breeds (although neither Labradoodle or Cavapoo significantly differed from their progenitor breeds regarding the odds of pruritus). Pruritus, often associated with skin disorders, is a commonly recorded dermatological condition in dogs, and other small domestic animals, in the UK [52, 62] and is often caused by allergies, including contact dermatitis and atopy. However, the current results are one of the first times pruritus has been recorded as a predisposition in a specific designer-crossbreed and is of relevance to overall UK dog health at a population level, considering Cockapoos were the second most popular dog breed aged under one year in the UK in 2019 [3]. All three designer-crossbreeds had higher odds of otitis externa than their non-Poodle progenitor breeds, although not compared to Poodles. Otitis externa is another disorder often triggered by allergies, and associated with environmental factors and the carriage of a dog's ears [40]. Designer-crossbreeds, Poodle-type breeds and Spaniel-type breeds have all previously been reported as predisposed to otitis externa, proposed in part as due to their tendencies towards pendulous ear carriage and hairy

ear canal [40]. Presence of abundant ear hair, as commonly seen in breeds like the Cocker Spaniel and Poodle, can cause retention of moisture and heat in the ear canal, increasing the risk of bacterial infection [63]. Otitis externa can carry serious implications for animal welfare, with this condition often requiring long-term treatment and frequent clinical visits [64]. The challenges faced by owners in managing otitis externa and other skin conditions may impair the human-dog bond and further compromise these dogs' welfare [62] as well as the owners' quality of life [65]. Diagnosis and monitoring of canine dermatological disorders is often limited due to its expensive, time-consuming and subjective nature, posing a challenge to veterinarians and owners alike [66]. Given that designer-crossbreeds with a Poodle progenitor often show distinctive curly coats, if it turns out that this coat type contributes to this predisposition, then breeding for these traits based on an aesthetic appeal may run counter to animal welfare.

Reduced odds were noted in a number of disorders in designer-crossbreeds. Both Labradoodle and Cockapoo had lower odds of patellar luxation compared to Poodles but did not differ from their other progenitor breed. Conversely, Cavapoo did not differ from either their CKCS or Poodle progenitor in their risk of patellar luxation. Previous studies have reported CKCS with higher odds of patellar luxation than crossbreeds, and that toy dog breeds, such as the Toy poodle, had increased odds of patellar luxation compared to larger dog breeds [67, 68]. Almost half (41.5%) of the Cavapoos in this study had Toy Poodle parentage, the smallest of the Poodle breeds. Given the shared risk factors of small size plus a purebred progenitor breed with high prevalence for patellar luxation, it is perhaps less surprising that Cavapoo did not show reduced odds of patellar luxation compared to the CKCS and Poodle progenitors, and thus did not show a potential positive hybrid vigour effect for this trait as did the Labradoodle and Cockapoo. This finding highlights the need for a stronger evidence base to support breeding programmes that aim to cross with known serious disorder predispositions.

The current findings on designer-crossbreed health can be used to support the design of many future research possibilities to better understand the health and welfare of this relatively new canine demographic. For example, future work could focus on identifying parentage and the generation number of cross to ascertain whether potential hybrid vigour effects have been obscured in research to date by sampling from populations that includes multi-generational crosses, rather than being restricted to F1 crosses, where these hybrid vigour effects are most likely to exist [69].

Additionally, age is known to strongly associate with the prevalence and likelihood of disorders in dogs [52]. All dogs, both designer-crossbreeds and their progenitor breeds, in the current study were relatively young (< 5yrs old), with age kept relatively consistent across the breeds to limit the effect of age upon health. However, this study design does mean that the health burden in more aged subsets of these emerging designer-crossbreeds remains unclear. A longitudinal study investigating disorder profiles of designer crossbreeds compared to their progenitor breeds as they age into their more senior years would be valuable to help owners know what to expect over time (e.g., move into the next canine age grouping [70]), and what preventative measures could be taken to slow and/or prevent these disease burdens.

It is important to note that the health results from this study relate directly only to the three designer-crossbreeds investigated (all of which were Poodle crosses) and should be extrapolated with caution to other designer-crossbreeds. Designer-crossbreeds that have a progenitor purebred with known health issues e.g. those associated with extreme conformations (e.g., brachycephalic breeds) or with genetic disorders with a recessive inherence pattern may still benefit from a hybrid vigour effect and exhibit relatively better health than that progenitor breed. The current study should be considered as introductory and further research into the health of other designer-crossbreeds is needed, particularly for breeding programmes where

the outcrossing is designed and motivated to counter known serious and common health issues of the purebred progenitors [71].

## Implications for current and future designer-crossbreed owners

The findings from the current study indicate that prospective owners should not use perceptions of enhanced health status as a reason to support acquisition of these three designer-crossbreeds. Better communication of the current evidence base on designer-crossbreed health is needed for both prospective and current owners of designer-crossbreed dogs, both pre- and post-purchase, to protect canine welfare in this population. This is particularly pertinent in this population because there is evidence that designer-crossbreed owners are less likely to follow important pre-purchasing practices (e.g., seeing the puppy with its mother) that risk acquisition from poorer welfare sources and may place these dogs at heightened risk of health problems [1]. The internet is now a key resource for prospective owners to source both information and also actual dogs [72] so good quality information on designer-crossbreed health should be made readily available via online resources, using user-friendly methods such as infographics and factsheets. Given that owners of non-pedigree and crossbreed dogs are more likely to source information from animal charities than owners of pedigree dogs [72] there is also a role here for leading canine charities to ensure readily available and good advice on designer-crossbreed health via their social media pages and websites. The increasing popularity of designer-crossbreeds means that this canine demographic is here to stay and many owners already consider these dogs as 'breeds' [73, 74]. Consequently, the time may now have arrived for kennel clubs to re-open their registers to designer-crossbreeds to play their part in collecting further health data, opening up the stud books, and providing detailed and evidence-based advice to prospective owners. Driving health improvements in dogs is known to be challenging [75]. However, increasing awareness of common health disorders in prospective owners of designer-crossbreeds may encourage consumer pressure on breeders to prioritise health over aesthetics when deciding which breeds and individual animals to cross (e.g., avoiding use of dogs affected by disorders in their breeding programme).

## Limitations of the study

The current study had some limitations, with attempts to mitigate taken where possible. The individual breed/crossbreed of dogs were not confirmed via pedigree history for registered purebreds or genetic sampling for designer crosses beyond the owners' certification. Some participants may not have known the precise parentage of their dog (particularly for designer-crossbreeds) or misinterpreted the crossbreed criteria (e.g., owners of Australian Labradoodles, who have Labrador Retriever, Poodle, English Cocker Spaniel, American Cocker Spaniel and Irish Water Spaniel parentage [76], may have believed they were eligible under the "Labradoodle" breed). Mitigation measures taken to limit these misclassifications included specifically defining Labradoodles as "Labrador Retriever crossed with any Poodle breed" in the survey instructions and the breed selection question. Owners have poorer dog health literacy compared to veterinary professionals, so therefore owner-reported health data may be less reliable compared to analysis of veterinary clinical records [59]. Future work on designer-crossbreed health could consider using parallel study designs that include both owner and veterinary clinical data collection. But even with veterinary clinical data, there are still challenges because some veterinary practices/practitioners still tend to use the generalised label of "crossbreed" for all crosses, designer or otherwise. Veterinary professionals and clinics could be encouraged to record more precise parentage (where known) in veterinary records so that clinical data on designer-crossbreeds can be used in further research (e.g., VetCompass studies) [48].

Due to the relatively lower Poodle numbers in the present study sample compared to the other pure breeds, the Poodle responses were merged so all three Poodle-type breeds were considered as a whole rather than as separate entities. This ensured that each study 'breed' had over 500 responses to facilitate higher powered statistical analyses. However, the health profiles of Toy, Miniature and Standard Poodles are not exactly the same so this may confound some of the current results [77, 78].

Although the questionnaire included a question asking what hybrid generation that the crossbreed was, the current analysis did not consider this variable. The main focus of this current health study compared between designer-crossbreed and progenitor purebred rather than a deeper exploration on the effects of various levels of hybrid generations. However, exploration of effects from the hybrid generation on disorder prevalence are important to consider and hopefully will be considered in a future analysis.

The use of convenience sampling methods can introduce selection bias because those owners who respond may not be fully representative of all owners. For example, it is possible that responding owners may be more welfare conscious and thus our results may represent a 'best case scenario' for all breeds studied because of data collected from owners who are more vigilant and proactive regarding their dog's health e.g., purchasing from health-focused breeders, using appropriate preventative healthcare [79]. Likewise, owners who have experienced problems and/or health issues in their dog in the past may have been more inclined to complete a questionnaire on their dog's health. To mitigate some of this selection bias, the current survey was promoted very broadly, including across many different social media platforms, breed and non-breed specific platforms, via diverse organisations with large owner/follower databases and also promoting on general pet or animal groups as a way of reaching a wider audience.

## Conclusions

The current results provide little support that the three designer-crossbreeds studied benefit from improved health due to crossbreeding, but instead exhibit overall health patterns that are equivalent largely with their purebred progenitor breeds. Given that perceived enhanced health is a major motivator of the acquisition of designer crossbreeds, communicating this message to prospective owners is of importance to align their expectations for future potential health issues and veterinary costs with reality and to ensure that any acquisition decisions are based on good evidence rather than marketing hype or social anecdote. Although this study provides a comprehensive and timely evaluation of the health of the UK's three most common designer-crossbreeds, given the growing popularity of Poodle-crosses [3], and paucity of research on this topic, further studies are warranted to test and expand the results reported.

## Supporting information

**S1 Text. Complete questionnaire text.**
(DOCX)

**S1 Fig. Recruitment posters.**
(DOCX)

**S1 Dataset. Health dataset.**
(XLSX)

## Acknowledgments

We would like to thank the following organisations for their support in disseminating the questionnaire: Pets4Homes, The Blue Cross, RSPCA. PDSA, VetPartners, Marc Abraham and the All-Parliamentary Dog Advisory Welfare Group (APDAWG), Pets at Home, BVA, Dog Breeding Reform Group (DBRG) and many other organisations and individuals. We would also like to express our sincere gratitude to all the dog owners who responded to the questionnaire.

## Author Contributions

**Data curation:** Gina T. Bryson.

**Formal analysis:** Gina T. Bryson.

**Methodology:** Gina T. Bryson.

**Resources:** Gina T. Bryson.

**Supervision:** Dan G. O'Neill, Claire L. Brand, Zoe Belshaw, Rowena M. A. Packer.

**Writing – original draft:** Gina T. Bryson.

**Writing – review & editing:** Gina T. Bryson.

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
