## [Decision Letter · Decision Letter 0]

5 Apr 2024

PONE-D-24-08401The Doodle Dilemma: How the physical health of ‘Designer-crossbreed’ Cockapoo, Labradoodle and Cavapoo dogs’ compares to their purebred progenitor breedsPLOS ONE

Dear Dr. Packer,

Thank you for submitting your manuscript to PLOS ONE. After careful consideration, we feel that it has merit but does not fully meet PLOS ONE’s publication criteria as it currently stands. Therefore, we invite you to submit a revised version of the manuscript that addresses the points raised during the review process. Please submit your revised manuscript by May 20 2024 11:59PM. If you will need more time than this to complete your revisions, please reply to this message or contact the journal office at plosone@plos.org. Please include the following items when submitting your revised manuscript:A rebuttal letter that responds to each point raised by the academic editor and reviewer(s). You should upload this letter as a separate file labeled 'Response to Reviewers'.A marked-up copy of your manuscript that highlights changes made to the original version. You should upload this as a separate file labeled 'Revised Manuscript with Track Changes'.An unmarked version of your revised paper without tracked changes. You should upload this as a separate file labeled 'Manuscript'.

We look forward to receiving your revised manuscript.

Kind regards,

Benito Soto-Blanco, DVM, MSc, PhD

Academic Editor

PLOS ONE

“R.MA.P and D.G.O(Kennel Club Charitable Trust) https://www.kennelclubcharitabletrust.org/

D.G.O (VetCompass)

https://www.vetcompass.org/”

“Firstly, I would like to thank the owners who responded to the questionnaire and all organisations and individuals who shared the questionnaire. I would also like to thank The Kennel Club Charitable Trust for helping to fund this project. Finally, a big thank you to my family who have been my biggest cheerleaders during this study.”

“R.MA.P and D.G.O(Kennel Club Charitable Trust) https://www.kennelclubcharitabletrust.org/

D.G.O (VetCompass)

https://www.vetcompass.org/”

6. Please include a separate caption for each figure in your manuscript.

Reviewers' comments:

Reviewer's Responses to Questions

**Comments to the Author**

1. Is the manuscript technically sound, and do the data support the conclusions?

Reviewer #1: Yes

Reviewer #2: Yes

2. Has the statistical analysis been performed appropriately and rigorously? 

Reviewer #1: Yes

Reviewer #2: I Don't Know

3. Have the authors made all data underlying the findings in their manuscript fully available?

Reviewer #1: Yes

Reviewer #2: No

4. Is the manuscript presented in an intelligible fashion and written in standard English?

Reviewer #1: Yes

Reviewer #2: Yes

5. Review Comments to the Author

Reviewer #1: The study is extremely valid and important as a source of accurate scientific information for veterinarians and owners. Because, through this study, there is enough sample data to show that the objective of hybrid vigor intended in these breeds simply does not exist. I reiterate that the objective of the study was achieved, as it fulfills the purpose of alerting veterinarians and owners that the acquisition of hybrid breeds is based on good evidence rather than marketing exaggerations or social jokes. I congratulate the authors and suggest that this research be extended to other canine hybrids.

Reviewer #2: Thank you for giving me the opportunity to review this very current and interesting study, which has the potential to aid not only prospective owners but also veterinarians in providing advice to potential owners. The article also makes a statement against unhealthy dog breeding.

Designer cross-breeds are increasingly popular not only in the UK. An important finding in this study is that they are neither healthier nor sicker than purebred dogs.

When kennel clubs close registers, not only in the UK, the possibility of out-crossing to increase genetic diversity is minimized. As stated in the manuscript, this is already a problem in certain breeds. Unfortunately, a majority of the diseases I, as a veterinarian, treat could have been avoided through healthy breeding practices.

General comments:

• Some references are difficult to find or contain missing information. For example, some references lack web addresses and access dates (e.g., numbers 55 and 69). Additionally, some references cannot be located at all (e.g., reference 26). Reference 30 appears to be a report of a scientific article and should refer to the original source. There also appears to be overlap between references, such as references 26 and 32, which seem to refer to the same project. Furthermore, some references do not seem to accurately represent the content they are referencing. For instance, in reference 53 the author explicitly states, 'The jury’s still out on that, but the general consensus is that mixed-breed dogs are no more or less likely to have health issues than their purebred counterparts.' However, this statement does not correspond to the sentence in the paper (line 515). While I do not doubt the accuracy of your statement, it is important to ensure that the references are correct to strengthen the argument. I have not checked all references, but when the first one I checked does not align, I suggest reviewing them to ensure they are used correctly.

• In general, I find that the discussion lacks comparisons with earlier studies, particularly regarding the prevalence of certain disorders among progenitor breeds. While some references are provided, the majority are from studies conducted by the same research group (Vet Compass data). I believe a more extensive comparison, supplemented with additional references, would add value and be more suitable.

• In the Results section, it is a bit unusual to report the chi-square value. I'm not sure what it adds; the reader would need the table and degrees of freedom to fully interpret it. I believe it is sufficient to report the p-value. However, I will leave this decision to someone more skilled in statistics.

Specific comments below:

Line 14: why is owner age and gender confounder? Suggest clarify (in M&M) (do some people report more often, seek veterinary advice?)

Line 29: …and health of parents?

Figure 1 in the introduction gives the impression that the designer crossbreeds (doodles) will soon be considered purebred breed. Very illustrative though, start one thinking!

370: The numbers don't add up to 100%. There may be incorrect numbers or missing answers. Please specify.

377: The numbers don’t make sense. Overall, n = 7,433 (79.0%) of the study population were insured. But in the two groups there were 88.3 % and 84.2% insured respectively?

381: Labradoodle, add percentage.

411: Change “higher odds” to “lower odds”. (OR is stated to be 0.56 and 0.41).

Table 2: Remove “Descriptive” , and “labrador” is missing after “vs”.

566-576: Allergy/allergic skin disorders, as well as alopecia, were lowest in the poodle progenitor group. Given this, is the discussion about atopy relevant in this paper? The references are thereby not relevant, suggest remove (61, 62, 63, 64, 65). To my knowledge, poodles are not predisposed to atopy. Since it appears that otitis externa is inherited from the poodle, perhaps focus on the hairy ear canals instead.

574-576: As mentioned above, since it is likely that hairy ear canals are inherited rather than atopy, it is suggested to rephrase the discussion accordingly.

592-598: Regarding the hybrid vigor effect. In the questionnaire, there was a question about which generation the dog belonged to, although this information is not reported in this manuscript. Could you please provide a rationale for this omission? Was there a distinction made between first-generation crosses and later generations? As far as I know, the heterosis effect is primarily observed in the first generation. While you briefly mention this in line 159, it would be valuable to clarify whether you included this distinction in your analyses.

626: health

679-680: Or the other way around, interest-bias; owners who chose to answer a questionnaire about their dogs' health are more prone to have experienced problems/health issues.

697: suggest remove “in the UK” as these crosses are common also in other western countries.

Supporting information: Add 'Do you own' to the file as: 'Cocker Spaniel?', to maintain consistency with the other breeds.

6. PLOS authors have the option to publish the peer review history of their article (what does this mean?). If published, this will include your full peer review and any attached files.

Reviewer #1: **Yes: **Paula Priscila Correia Costa

Reviewer #2: **Yes: **Karolina Brunius Enlund

---

## [Author Response · Author response to Decision Letter 0]

4 Jun 2024

Reviewers' comments:

5. Review Comments to the Author

Reviewer #1: 

C: The study is extremely valid and important as a source of accurate scientific information for veterinarians and owners. Because, through this study, there is enough sample data to show that the objective of hybrid vigor intended in these breeds simply does not exist. I reiterate that the objective of the study was achieved, as it fulfils the purpose of alerting veterinarians and owners that the acquisition of hybrid breeds is based on good evidence rather than marketing exaggerations or social jokes. I congratulate the authors and suggest that this research be extended to other canine hybrids.

R: Thank you so much for your kind words and for generously giving up your time to review this paper. We are delighted that you found the content informative and you were satisfied the main objective had been achieved.

Reviewer #2: 

C: Thank you for giving me the opportunity to review this very current and interesting study, which has the potential to aid not only prospective owners but also veterinarians in providing advice to potential owners. The article also makes a statement against unhealthy dog breeding.

Designer cross-breeds are increasingly popular not only in the UK. An important finding in this study is that they are neither healthier nor sicker than purebred dogs.

When kennel clubs close registers, not only in the UK, the possibility of out-crossing to increase genetic diversity is minimized. As stated in the manuscript, this is already a problem in certain breeds. Unfortunately, a majority of the diseases I, as a veterinarian, treat could have been avoided through healthy breeding practices.

R: Thank you for reviewing this paper, we are pleased you find it relevant and beneficial to current dog owners. It is encouraging to hear how the content of this paper has the potential to be of use in your profession. We have read through your following comments carefully and hope that you find the alterations suitable.

General comments:

C: Some references are difficult to find or contain missing information. For example, some references lack web addresses and access dates (e.g., numbers 55 and 69). Additionally, some references cannot be located at all (e.g., reference 26). Reference 30 appears to be a report of a scientific article and should refer to the original source. There also appears to be overlap between references, such as references 26 and 32, which seem to refer to the same project. Furthermore, some references do not seem to accurately represent the content they are referencing. For instance, in reference 53 the author explicitly states, 'The jury’s still out on that, but the general consensus is that mixed-breed dogs are no more or less likely to have health issues than their purebred counterparts.' However, this statement does not correspond to the sentence in the paper (line 515). While I do not doubt the accuracy of your statement, it is important to ensure that the references are correct to strengthen the argument. I have not checked all references, but when the first one I checked does not align, I suggest reviewing them to ensure they are used correctly.

R: Thank you for your observation. We have gone through each individual reference and made sure it is in the correct format and includes all necessary information. For a few of the journal articles page numbers are not given however we have made sure volume and date are present. We have removed the reference 26 and replaced with 32 as they were the same article. Regarding the reference 53, we have replaced this with a more relevant article that focuses more on crossbreed bias and supports our argument in the paper. We hope you find this an acceptable action and any further queries with referencing will be picked up during final formatting by the editorial team. 

C: • In general, I find that the discussion lacks comparisons with earlier studies, particularly regarding the prevalence of certain disorders among progenitor breeds. While some references are provided, the majority are from studies conducted by the same research group (Vet Compass data). I believe a more extensive comparison, supplemented with additional references, would add value and be more suitable.

R: Thank you for your comments on the discussion, we appreciate that a lot of the studies we reference are indeed VetCompass data. This is because these are often large-scale canine health studies that utilise data from veterinary records across the UK and consequently, provide findings with high validity. Having said this, we do understand that the discussion could benefit from having alternative references from other canine studies and we have therefore ensured to include numerous references from other sources that help to reinforce our overall message in the discussion. Examples of some of the discussion changes and addition referencing:

Lines 665 – 666: Numerous studies exploring health and longevity in UK dogs reveal enteropathy as a highly prevalent canine disorder in the UK [52,57**, 58]

*Refs 57 [Adams V, Evans K, Sampson, J, Wood, J. Methods and mortality results of a health survey of purebred dogs in the UK. The Journal of small animal practice. 2010; 51(10): 512-24.]

**58 [Kathrani A, Werling D, Allenspach K. Canine breeds at high risk of developing inflammatory bowel disease in the south-eastern UK. Vet Rec. 2011;169(24):635]

Line 681 – 684: Pruritus, often associated with skin disorders, is a commonly recorded dermatological condition in dogs, and other small domestic animals, in the UK [52, 62*] and is often caused by allergies, including contact dermatitis and atopy.

Ref 62* [Hill, P.B., Lo, A., Eden, C.A.N., Huntley, S., Morey, V., Ramsey, S., Richardson, C., Smith, D.J., Sutton, C., Taylor, M.D., Thorpe, E., Tidmarsh, R. and Williams, V. Survey of the prevalence, diagnosis and treatment of dermatological conditions in small animals in general practice. Veterinary Record, 2006; 158 (16): 533-539]

Line 699-701: Presence of abundant ear hair, as commonly seen in breeds like the Cocker Spaniel and Poodle, can cause retention of moisture and heat in the ear canal, increasing the risk of bacterial infection [64*]

*Ref 66 [Hayes HM Jr, Pickle LW, Wilson GP. Effects of ear type and weather on the hospital prevalence of canine otitis externa. Res Vet Sci. 1987;42(3):294-298]

Lines: 715-717: Previous studies have reported CKCS with higher odds of patellar luxation than crossbreeds, and that toy dog breeds, such as the Toy poodle, had increased odds of patellar luxation compared to larger dog breeds [68, 69*].

*Ref 69 [Priester WA. Sex, size and breed as risk factors in canine patellar dislocation. Journal of the American Veterinary Medication Association. 1973; 160(5): 740-742]

C: In the Results section, it is a bit unusual to report the chi-square value. I'm not sure what it adds; the reader would need the table and degrees of freedom to fully interpret it. I believe it is sufficient to report the p-value. However, I will leave this decision to someone more skilled in statistics.

R: Thank you for this, you’re right that the reader would ideally need degrees of freedom to fully comprehend the statistic results so we have gone through and removed all Chi-square values, just leaving the p-value.

Specific comments below:

C: Line 14: why is owner age and gender confounder? Suggest clarify (in M&M) (do some people report more often, seek veterinary advice?)

R: This is a great question, both owner age and gender have been noted in earlier canine health studies as confounding factors that influence how owners report their dog’s health. For example, a study conducted in Sweden investigating owner’s perspectives of dog’s dental health indicated that men perceive dental health in their dog better than women dog owners. The study also revealed that younger dog owners believed the dental health of their dogs to be worse than older dog owners. I have made sure to include these specific study references in the methodology.

Lines 330-334: Each multivariable model included a fixed set of additional variables to account for confounding which were selected using an information theory approach [51] developed from other canine health studies that identified the following as confounding factors [52, 53] : dog age, sex, neuter status, insured status; and owner gender and age [45, 54]. Statistical significance was set at p < 0.05.

C: Line 29: …and health of parents?

R: That’s a good point to add thank you

Line 36-37:….on other factors important to canine welfare such as breeding conditions, conformation and health of parents.

C: Figure 1 in the introduction gives the impression that the designer crossbreeds (doodles) will soon be considered purebred breed. Very illustrative though, start one thinking!

R: Thank you we are glad that you find the figure of interest, you are right that it definitely gets you thinking. Many purebreds we see today originated as a form of designer-crossbreed. The designer-crossbreeds we see today may even be considered purebred in a few generations time. You raise a very good point and this is something we have now included in the discussion.

Lines 625-628: …This is particularly pertinent when referring back to figure 1 which illustrates how many of today’s purebreds originated from crossbreeding between distinct purebred breeds and were therefore designer-crossbreeds themselves during the early phase of their purebred development.

C: 370: The numbers don't add up to 100%. There may be incorrect numbers or missing answers. Please specify

R: Thank you for noticing this, we originally omitted detail about the dogs with an unspecified neuter status in order to save on words, however we understand that this is still valuable to include alongside the rest of the neuter statistics. We have therefore added the correct percentage and number of dogs with unspecified neuter status.

Line 429-430: The overall study population included n = 4,786 (50.9%) entire dogs and n = 3,869 (41.1%) neutered dogs. The neuter status of n = 747 (8%) dogs was unspecified.

C: 377: The numbers don’t make sense. Overall, n = 7,433 (79.0%) of the study population were insured. But in the two groups there were 88.3 % and 84.2% insured respectively? 381: Labradoodle, add percentage.

R: Thank you for spotting this mistake, we have re-done the statistics correctly so that it now makes sense and shows the right percentages. We have also added the percentage for the Labradoodle.

Line 436-453: Overall, n = 7,432 (79.0%) of the study population were insured. Designer-crossbreeds were more likely to be insured than purebreds (insured: designer-crossbreeds n = 2,826 (82.5%) purebreds n = 4,606 (77.1%); X2: 26.51, p<0.001). The probability of being insured did not differ between the three designer-crossbreeds, (Cockapoo n = 1,535, 88.4%, Cavapoo n = 803, 87.6%, Labradoodle n = 489, 89.6%, X2: 1.32, p = 0.516).

C: 411: Change “higher odds” to “lower odds”. (OR is stated to be 0.56 and 0.41).

Table 2: Remove “Descriptive” , and “labrador” is missing after “vs”

R: We have updated this section so that the correct odds now show for the Cockapoo vs Cocker Spaniel so that it now states the correct odds values. We have removed the word “descriptive” from the table however, we cannot find the missing “labrador” after the vs?

Lines 494: Compared to their Cocker Spaniel progenitor breed, Cockapoos had lower odds… 

C: 566-576: Allergy/allergic skin disorders, as well as alopecia, were lowest in the poodle progenitor group. Given this, is the discussion about atopy relevant in this paper? The references are thereby not relevant, suggest remove (61, 62, 63, 64, 65). To my knowledge, poodles are not predisposed to atopy. Since it appears that otitis externa is inherited from the poodle, perhaps focus on the hairy ear canals instead.

C: 574-576: As mentioned above, since it is likely that hairy ear canals are inherited rather than atopy, it is suggested to rephrase the discussion accordingly.

R: Thank you for these insightful suggestions, you are correct that Poodles do not appear to be predisposed to atopy and that consequently, it would be much better to include references focused more on otitis externa and the factors that increase the likelihood of this condition. We have subsequently altered this section in the discussion to incorporate your valuable advice. 

Lines 697 – 710: Designer-crossbreeds, Poodle-type breeds and Spaniel-type breeds have all previously been reported as predisposed to otitis externa, proposed in part as due to their tendencies towards pendulous ear carriage and hairy ear canal [42]. Presence of abundant ear hair, as commonly seen in breeds like the Cocker Spaniel and Poodle, can cause retention of moisture and heat in the ear canal, increasing the risk of bacterial infection [61]. Otitis externa can carry serious implications for animal welfare, with this condition often requiring long-term treatment and frequent clinical visits [62]. The challenges faced by owners in managing otitis externa and other skin conditions may impair the human-dog bond and further compromise these dogs’ welfare [62] as well as the owners’ quality of life [63]. Diagnosis and monitoring of canine dermatological disorders is often limited due to its expensive, time-consuming and subjective nature, posing a challenge to veterinarians and owners alike [64]. Given that designer-crossbreeds with a Poodle progenitor often show distinctive curly coats, if it turns out that this coat type contributes to this predisposition, then breeding for these traits based on an aesthetic appeal may run counter to animal welfare. 

C: 592-598: Regarding the hybrid vigor effect. In the questionnaire, there was a question about which generation the dog belonged to, although this information is not reported in this manuscript. Could you please provide a rationale for this omission? Was there a distinction made between first-generation crosses and later generations? As far as I know, the heterosis effect is primarily observed in the first generation. While you briefly mention this in line 159, it would be valuable to clarify whether you included this distinction in your analyses.

R: We thank you for your query and understand that further clarification is needed. You are correct that in the questionnaire, we asked owners what generation their crossbreed was. However, due to the scale of the project and the fact we were restricted to a tight timeline (this was an MRes study which was only 1 year) we prioritised the key aim of comparing health between progenitor and designer-crossbreed, regardless of their generation. However, we recognise the importance of this information and believe it would be valuable to explore further. We hope to use the data from the questionnaire, as well as future RVC/VetCompass work, to investigate the effect of generation type on canine health over the next two years. We have made sure to include this reasoning in the limitations of the paper. 

Lines 849 – 854: Although the questionnaire included a question asking what hybrid generation that the crossbreed was, the current analysis did not consider this variable. The main focus of this current health study compared between designer-crossbreed and progenitor purebred rather than a deeper exploration on the effects of various levels of hybrid generations. However, exploration of effects from the hybrid generation on disorder prevalence are important to consider and hopefully will be considered in a future analysis. 

C: 626: health

R: Thank you for spotting this. I have corrected it. 

Line 787: Better communication of the current evidence base on designer-crossbreed health is needed for both prospective and current owners of designer-crossbreed dogs, both pre- and post-purchase, to protect canine welfare in this population.

C: 679-680: Or the other way around, interest-bias; owners who chose to answer a questionnaire about their dogs' health are more prone to have experienced problems/health issues.

R: Very true, I am aware that convenience sampling introduces bi

---

## [Decision Letter · Decision Letter 1]

16 Jun 2024

The Doodle Dilemma: How the physical health of ‘Designer-crossbreed’ Cockapoo, Labradoodle and Cavapoo dogs’ compares to their purebred progenitor breeds

PONE-D-24-08401R1

Dear Dr. Packer,

We’re pleased to inform you that your manuscript has been judged scientifically suitable for publication and will be formally accepted for publication once it meets all outstanding technical requirements.

Kind regards,

Benito Soto-Blanco, DVM, MSc, PhD

Academic Editor

PLOS ONE

Reviewers' comments:

Reviewer's Responses to Questions

**Comments to the Author**

1. If the authors have adequately addressed your comments raised in a previous round of review and you feel that this manuscript is now acceptable for publication, you may indicate that here to bypass the “Comments to the Author” section, enter your conflict of interest statement in the “Confidential to Editor” section, and submit your "Accept" recommendation.

Reviewer #2: All comments have been addressed

2. Is the manuscript technically sound, and do the data support the conclusions?

Reviewer #2: Yes

3. Has the statistical analysis been performed appropriately and rigorously? 

Reviewer #2: I Don't Know

4. Have the authors made all data underlying the findings in their manuscript fully available?

Reviewer #2: No

5. Is the manuscript presented in an intelligible fashion and written in standard English?

Reviewer #2: Yes

6. Review Comments to the Author

Reviewer #2: Thank you for addressing all concerns. I´m still not sure about the numbers of insured dogs though:). The insured designer-breeds were 82.5%, however when specified they are 87.6, 88.4 and 89.6 % respectively?

Regarding underlying data availability, there are usually restrictions due to confidentiality to share all gathered questionnaire data, and I believe that it is acceptable as it is in this study. Maybe anonymized raw data is available in some repository although I have not seen it. I leave this to the editor.

7. PLOS authors have the option to publish the peer review history of their article (what does this mean?). If published, this will include your full peer review and any attached files.

Reviewer #2: **Yes: **Karolina Brunius Enlund

---

## [Editor Report · Acceptance letter]

2 Aug 2024

PONE-D-24-08401R1 

PLOS ONE

Dear Dr. Packer, 

I'm pleased to inform you that your manuscript has been deemed suitable for publication in PLOS ONE. Congratulations! Your manuscript is now being handed over to our production team.

Kind regards, 

on behalf of

Dr. Benito Soto-Blanco 

Academic Editor

PLOS ONE